# FINE-TUNING DISCRETE DIFFUSION MODELS VIA REWARD OPTIMIZATION WITH APPLICATIONS TO DNA AND PROTEIN DESIGN

**Chenyu Wang**[*†1], **Masatoshi Uehara**[*2], **Yichun He**[†3], **Amy Wang**[2], **Avantika Lal**[2],

**Tommi Jaakkola**[1], **Sergey Levine**[4], **Aviv Regev**[2], **Hanchen Wang**[‡2,5], **Tommaso Biancalani**[‡2]

[1]MIT  [2]Genentech  [3]Harvard University  [4]UC Berkeley  [5]Stanford University

## ABSTRACT

Recent studies have demonstrated the strong empirical performance of diffusion models on discrete sequences (i.e., discrete diffusion models) across domains from natural language to biological sequence generation. For example, in the protein inverse folding task, where the goal is to generate a protein sequence from a given backbone structure, conditional diffusion models have achieved impressive results in generating natural-like sequences that fold back into the original structure. However, practical design tasks often require not only modeling a conditional distribution but also optimizing specific task objectives. For instance, in the inverse folding task, we may prefer protein sequences with high stability. To address this, we consider the scenario where we have pre-trained discrete diffusion models that can generate natural-like sequences, as well as reward models that map sequences to task objectives. We then formulate the reward maximization problem within discrete diffusion models, analogous to reinforcement learning (RL), while minimizing the KL divergence against pretrained diffusion models to preserve naturalness. To solve this RL problem, we propose a novel algorithm, **DRAKES**, that enables direct backpropagation of rewards through entire trajectories generated by diffusion models, by making the originally non-differentiable trajectories differentiable using the Gumbel-Softmax trick. Our theoretical analysis indicates that our approach can generate sequences that are both natural-like (i.e., have a high probability under a pretrained model) and yield high rewards. While similar tasks have been recently explored in diffusion models for continuous domains, our work addresses unique algorithmic and theoretical challenges specific to discrete diffusion models, which arise from their foundation in continuous-time Markov chains rather than Brownian motion. Finally, we demonstrate the effectiveness of our algorithm in generating DNA and protein sequences that optimize enhancer activity and protein stability, respectively, important tasks for gene therapies and protein-based therapeutics. The code is available at `https://github.com/ChenyuWang-Monica/DRAKES`.

## 1 INTRODUCTION

Diffusion models have gained widespread recognition as effective generative models in continuous spaces, such as image and video generation (Song et al., 2020a; Ho et al., 2022). Inspired by seminal works (e.g., Austin et al. (2021); Campbell et al. (2022); Sun et al. (2022)), recent studies (Lou et al., 2023; Shi et al., 2024b; Sahoo et al., 2024) have shown that diffusion models are also highly effective in discrete spaces, including natural language and biological sequence generation (DNA, RNA, proteins). Unlike autoregressive models commonly used in language modeling, diffusion models are particularly well-suited for biological sequences, where long-range interactions are crucial

---

[*]Equal contribution: wangchy@mit.edu, uehara.masatoshi@gene.com

[†]Work mainly done during an internship at Genentech.

[‡]Corresponding authors: wang.hanchen@gene.com, biancalt@gene.com

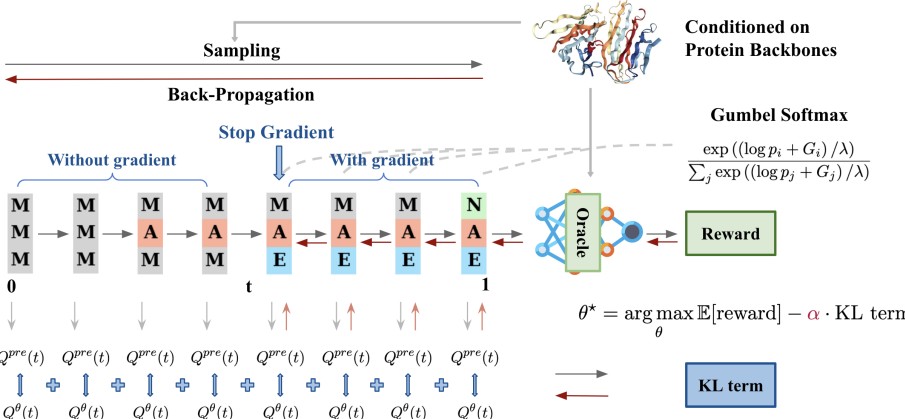

Figure 1: **DRAKES**. We maximize the reward with a penalty term relative to pre-trained discrete diffusion models using the Gumbel-Softmax trick.

for the physical behavior of molecules arising from those sequences (e.g., the 3D folded structure of RNA or proteins).

While discrete diffusion models effectively capture conditional distributions (e.g., the distribution of sequences given a specific backbone structure in an inverse protein folding design problem (Dauparas et al., 2022; Campbell et al., 2024)), in many applications, especially for therapeutic discovery, we often aim to generate sequences that are both natural-like and optimize a downstream performance objective. For instance, in the inverse folding problem, we may prefer stable protein sequences (i.e., sequences that fold back into stable protein conformations (Widatalla et al., 2024)); for mRNA vaccine production we desire 5' UTRs that drive high translational efficiency (Castillo-Hair & Seelig, 2021); for gene and cell therapies, we desire regulatory DNA elements, such as promoters and enhancers, that drive high gene expression only in specific cell types (Taskiran et al., 2024); and for natural language we optimize to minimize harmfulness (Touvron et al., 2023).

To address these challenges, our work introduces a fine-tuning approach for well-pretrained discrete diffusion models that maximizes downstream reward functions. Specifically, we aim to optimize these reward functions while ensuring that the generated sequences maintain a high probability under the original conditional distribution (e.g., the distribution of sequences that fold into a given backbone structure). To achieve this, we formulate the problem as a reward maximization task, analogous to reinforcement learning (RL), where the objective function integrates both the reward terms and the KL divergence with respect to the pre-trained discrete diffusion model, which ensures that the generated sequences remain close to the pre-trained model, preserving their naturalness after fine-tuning. To solve this RL problem, we propose a novel algorithm, **DRAKES**, that enables direct backpropagation of rewards through entire trajectories by making the originally non-differentiable trajectories differentiable using the Gumbel-Softmax trick (Jang et al., 2016; Maddison et al., 2016).

Our main contribution is an RL-based fine-tuning algorithm, Direct Reward bAcKpropagation with gumbEl Softmax trick (**DRAKES**), that enables reward-maximizing finetuning for discrete diffusion models (Figure 1). We derive a theoretical guarantee that demonstrates its ability to generate *natural* and *high-reward* designs, and demonstrate its performance empirically on DNA and protein design tasks. While similar algorithms exist for continuous spaces (Fan et al., 2023; Black et al., 2023; Uehara et al., 2024b; Venkatraman et al., 2024; Yuan et al., 2023; Guo et al., 2024), our work is the first, to the best of our knowledge, to address these aspects in (continuous-time) discrete diffusion models. This requires addressing unique challenges, as discrete diffusion models are formulated as continuous-time Markov chains (CTMC), which differ from Brownian motion, and the induced trajectories from CTMC are no longer differentiable, unlike in continuous spaces. Our novel theoretical guarantee also establishes a connection with recent advancements in classifier guidance for discrete diffusion models (Nisonoff et al., 2024).

## 2 RELATED WORKS

**Discrete diffusion models and their application in biology.** Building on the seminal works of Austin et al. (2021); Campbell et al. (2022), recent studies on masked diffusion models (Lou et al.,

2023; Shi et al., 2024b; Sahoo et al., 2024) have demonstrated strong performance in natural language generation. Recent advances in masked discrete diffusion models have been successfully applied to biological sequence generation, including DNA and protein sequences (Sarkar et al., 2024; Campbell et al., 2024). Compared to autoregressive models, diffusion models may be particularly well-suited for biological sequences, which typically yield molecules that fold into complex three-dimensional (3D) structures. In contrast to these works, our study focuses on fine-tuning diffusion models to optimize downstream reward functions. One application of our approach is the fine-tuning of protein inverse folding generative models to optimize stability, as discussed in Widatalla et al. (2024). However, unlike this prior work, we employ discrete diffusion models as the generative model.

**Controlled generation in diffusion models.** There are three primary approaches:

- **Guidance**: Techniques such as classifier guidance (Song et al., 2020b; Dhariwal & Nichol, 2021) and its variants (e.g., Bansal et al. (2023); Chung et al. (2022a); Ho et al. (2022)) introduce gradients from proxy models during inference. However, since gradients are not formally well-defined for discrete states in diffusion, a recent study (Nisonoff et al., 2024) proposed a method specifically designed for discrete diffusion models. Alternative approaches directly applicable to discrete diffusion models include sequential Monte Carlo (SMC)-based methods (Wu et al., 2024; Trippe et al., 2022; Dou & Song, 2024; Cardoso et al., 2023; Phillips et al., 2024). While these guidance-based inference techniques have their own advantages, they generally lead to longer inference times compared to fine-tuned models. We compare our methods against these in terms of generation quality in Section 6.

- **RL-based fine-tuning**: To maximize reward functions for pretrained diffusion models, numerous recent studies have explored RL-based fine-tuning in continuous diffusion models (i.e., diffusion models for continuous objectives) (Fan et al., 2023; Black et al., 2023; Clark et al., 2023; Prabhudesai et al., 2023). Our work, in contrast, focuses on discrete diffusion models.

- **Classifier-free fine-tuning (Ho & Salimans, 2022)**: This approach constructs conditional generative models, applicable in our setting by conditioning on high reward values. Although not originally designed as a fine-tuning method, it can also be adapted for fine-tuning (Zhang et al., 2023) by adding further controls to optimize. However, in the context of continuous diffusion models, compared to RL-based fine-tuning, several works (Uehara et al., 2024c) have shown that conditioning on high reward values is suboptimal, because such high-reward samples are rare. We will likewise compare this approach to ours in Section 6. Lastly, when pretrained models are conditional diffusion models (i.e., $p(x|c)$) and the offline dataset size consisting of triplets $(c, x, r(x))$ is limited, it is challenging to achieve success. Indeed, for this reason, most current RL-based fine-tuning papers (e.g., Fan et al. (2023); Black et al. (2023); Clark et al. (2023)) do not empirically compare their algorithms with classifier-free guidance.

## 3 PRELIMINARY

### 3.1 DIFFUSION MODELS ON DISCRETE SPACES

In diffusion models, our goal is to model the data distribution $p_{\text{data}} \in \Delta(\mathcal{X})$ using the training data, where $\mathcal{X}$ represents the domain. We focus on the case where $\mathcal{X} = \{1, 2, \cdots, N\}$. The fundamental principle is (1) introducing a known forward model that maps the data distribution to a noise distribution, and (2) learning the time reversal that maps the noise distribution back to the data distribution (detailed in Lou et al. (2023); Sahoo et al. (2024); Shi et al. (2024a)).

First, we consider the family of distributions $j_t \in \mathbb{R}^N$ (a vector summing to 1) that evolves from $t = 0$ to $t = T$ according to a continuous-time Markov chain (CTMC):

$$\frac{dj_t}{dt} = Q(t)j_t, \quad p_0 \sim p_{\text{data}}, \tag{1}$$

where $Q(t) \in \mathbb{R}^{N \times N}$ is the generator. Generally, $j_t$ is designed so that $p_t$ approaches a simple limiting distribution at $t = T$. A common approach is to add *Mask* into $\mathcal{X}$ and gradually mask a sequence so that the limiting distribution becomes completely masked (Shi et al., 2024a; Sahoo et al., 2024).

Next, we consider the time-reversal CTMC (Sun et al., 2022) that preserves the marginal distribution. This can be expressed as follows:

$$\frac{dj_{T-t}}{dt} = \bar{Q}(T-t)j_{T-t}, \quad \bar{Q}_{x,y}(t) = \begin{cases} \frac{j_t(y)}{j_t(x)} Q_{y,x}(t) \, (y \neq x) \\ -\sum_{y \neq x} \bar{Q}_{x,y}(t) \, (y = x), \end{cases} \tag{2}$$

where $Q_{x,y}(t)$ is a $(x, y)$-entry of a generator $Q(t)$, representing the transition rate matrix from state $x$ to state $y$. This implies that if we can learn the marginal density ratio $p_t(y)/p_t(x)$, we can sample from the data distribution at $t = T$ by following the above CTMC controlled by $\bar{Q}(T - t)$. Existing works (e.g., Lou et al. (2023)) demonstrate how to train this ratio from the training data. Especially when we use masked diffusion models (Sahoo et al., 2024; Shi et al., 2024a), we get

$$\bar{Q}_{x,y}(t) = \begin{cases} \gamma \mathbb{E}[x_0 = y | x_t = \text{Mask}] & (y \neq \text{Mask}, x_t = \text{Mask}), \\ -\sum_{z \neq \text{Mask}} \gamma \mathbb{E}[x_0 = z | x_t = \text{Mask}] & (y = \text{Mask}, x_t = \text{Mask}) \\ 0 & (x_t \neq \text{Mask}) \end{cases}, \tag{3}$$

for a certain constant $\gamma$, where the expectation is taken with respect to (w.r.t.) the distribution induced by the forward CTMC. Notably, the above formulation suggests that masked diffusion models could be viewed as a hierarchical extension of BERT (Devlin, 2018).

**Remark 1** (Sequence of multiple tokens). *When dealing with sequences of length $M$, $x = [x^{\langle 1 \rangle}, \cdots, x^{\langle M \rangle}]$, we simply consider the factorized rate matrix, i.e., $\bar{Q}_{x,y} = \sum_i \bar{Q}_{x,y^{\langle i \rangle}}$ (Campbell et al., 2022), thereby avoiding exponential blowup.*

**Remark 2** (Conditioning). *We can easily construct a conditional generative model for any $c \in \mathcal{C}$ by allowing the generator to be a function of $c \in \mathcal{C}$.*

## 3.2 GOAL: GENERATING NATURAL SAMPLES WHILE OPTIMIZING REWARD FUNCTIONS

In our work, we consider a scenario with a pretrained masked discrete diffusion model $p^{\text{pre}}(x|c) \in [\mathcal{C} \rightarrow \Delta(\mathcal{X})]$ trained on an extensive dataset and a downstream reward function $r : \mathcal{X} \rightarrow \mathbb{R}$. The pretrained diffusion model captures the naturalness or validity of samples. For example, in protein design, $p^{\text{pre}}(\cdot|\cdot)$ could be a protein inverse-folding model that generates amino acid sequences that fold back into the given backbone structure (similar to Campbell et al. (2024)), and $r$ could be a function that evaluates stability. Our objective is to fine-tune a generative model to generate *natural-like* samples (high $\log p^{\text{pre}}(\cdot|\cdot)$) with desirable properties (high $r(\cdot)$).

**Notation.** We introduce a discrete diffusion model parameterized by $\theta$ from $t = 0$ to $t = T$[1]:

$$\frac{dp_t}{dt} = Q^\theta(t)p_t, \quad p_0 = p_{\text{lim}}. \tag{4}$$

The parameter $\theta$ from the pretrained model is denoted by $\theta_{\text{pre}}$ and $p_{\text{lim}}$ denotes the initial distribution. The distribution at time $T$ is denoted as $p^{\text{pre}}(\cdot)$, which approximates the training data distribution $p^{\text{data}}$. We denote an element of the generated trajectory from $t = 0$ to $t = T$ by $x_{0:T}$. For simplicity, we assume the initial distribution is a Dirac delta distribution (completely masked state), and we often treat the original pretrained diffusion model as an unconditional model for a single token for notational convenience. In this paper, all of the proofs are in Appendix C. We provide an extension to the case when the initial distributions are stochastic in Appendix D.

## 4 ALGORITHM

In this section, we present our proposed method, **DRAKES**, for fine-tuning diffusion models to optimize downstream reward functions. We begin by discussing the motivation behind our algorithm.

## 4.1 KEY FORMULATION

Perhaps the most obvious starting point for fine-tuning diffusion models to maximize a reward function $r(x_T)$ is to simply maximize the expected value of the reward under the model's distribution, i.e., $\mathbb{E}_{x_0 \sim P^\theta}[r(x_T)]$, where the expectation is taken over the distribution $P^\theta(x_{0:T})$ induced by (4) (i.e., the generator $Q^\theta$). However, using only this objective could lead to over-optimization, where the model produces unrealistic or unnatural samples that technically achieve a high reward, but are impossible to generate in reality. Such samples typically exploit flaws in the reward function, for example, by being outside the training distribution of a learned reward or violating the physical assumptions of a hand-engineered physics-based reward (Levine et al., 2020; Clark et al., 2023; Uehara et al., 2024c). We address this challenge by constraining the optimized model to remain close to a pretrained diffusion model, which captures the distribution over natural or realistic samples. More specifically, we introduce a penalization term by incorporating the KL divergence between the fine-tuned model $P^\theta(x_{0:T})$ and the pretrained diffusion model $P^{\theta_{\text{pre}}}(x_{0:T})$ in CTMC.

---

[1]Starting from Section 3.2, to simply the notation, we go from $t = 0$ to $t = T$ to represent noise to data.

Accordingly, our goal during fine-tuning is to solve the following reinforcement learning (RL) problem:

$$\theta^\star = \underset{\theta \in \Theta}{\operatorname{argmax}} \underbrace{\mathbb{E}_{x_{0:T} \sim P^\theta}[r(x_T)]}_{\text{Reward term}} \tag{5}$$

$$- \alpha \underbrace{\mathbb{E}_{x_{0:T} \sim P^\theta}\left[\int_{t=0}^T \sum_{y \neq x_t} \left\{ Q_{x_t,y}^{\theta^{\text{pre}}}(t) - Q_{x_t,y}^\theta(t) + Q_{x_t,y}^\theta(t) \log \frac{Q_{x_t,y}^\theta(t)}{Q_{x_t,y}^{\theta^{\text{pre}}}(t)} \right\} dt \right]}_{\text{KL term}}.$$

The first term is designed to generate samples with desired properties, while the second term represents the KL divergence. The parameter $\alpha$ controls the strength of this regularization term.

Finally, after fine-tuning, by using the following CTMC from $t = 0$ to $t = T$:

$$\tfrac{dp_t}{dt} = Q^{\theta^\star}(t)p_t, \quad p_0 = p_{\lim}. \tag{6}$$

we generate samples at time $T$. Interestingly, we can show the following.

**Theorem 1** (Fine-Tuned Distribution). *When $\{Q_{\cdot,\cdot}^\theta : \theta \in \Theta\}$ is fully nonparametric (i.e., realizability holds), the generated distribution at time $T$ by* (6) *is proportional to*

$$\exp(r(\cdot)/\alpha)p^{\text{pre}}(\cdot). \tag{7}$$

This theorem offers valuable insights. The first term, $\exp(r(x))$, represents high rewards. Additionally, the second term, $p^{\text{pre}}(\cdot)$, can be seen as prior information that characterizes the natural sequence. For example, in the context of inverse protein folding, this refers to the ability to fold back into the target backbone structure.

**Remark 3.** *A similar theorem has been derived for continuous diffusion models (Uehara et al., 2024b, Theorem 1). However, our formulation* (5) *differs significantly as our framework is based on a CTMC, whereas those works are centered around the Brownian motion. Furthermore, while the use of a similar distribution is common in the literature on (autoregressive) large language models (e.g., Ziegler et al. (2019)), its application in discrete diffusion models is novel, considering that $p^{\text{pre}}(\cdot)$ cannot be explicitly obtained in our context, unlike autoregressive models.*

## 4.2 DIRECT REWARD BACKPROPAGATION WITH GUMBEL SOFTMAX TRICK (**DRAKES**)

Based on the key formulation presented in Section 4.1, we introduce our proposed method (Algorithm 1 and Figure 1), which is designed to solve the RL problem (5). The core approach involves iteratively (a) sampling from $x_{0:T} \sim P^\theta$ and (b) updating $\theta$ by approximating the objective function (5) with its empirical counterpart and adding its gradient with respect to $\theta$ into the current $\theta$. Importantly, for step (b) to be valid, step (a) must retain the gradients from $\theta$. After explaining the representation of $x_t$, we will provide details on each step.

**Representation.** To represent $x \in \{1, \cdots, N\}$, we often use the $N$-dimensional one-hot encoding representation within $\mathbb{R}^N$ interchangeably. From this perspective, while the original generator corresponds to a map $\mathcal{X} \times \mathcal{X} \to \mathbb{R}$, we can also regard it as an extended mapping: $\mathbb{R}^N \times \mathbb{R}^N \to \mathbb{R}$. We will use this extended mapping when we consider our algorithm later.

**Stage 1: Data collection (Step 2-9)** We aim to sample from the distribution induced by the generator $Q^\theta$. In the standard discretization of CTMC, for $(y, x) \in \mathcal{X} \times \mathcal{X}$, at time $t$, we use

$$p(x_{t+\Delta t} = y | x_t = x) = \text{I}(x = y) + Q_{x,y}^\theta(t)(\Delta t). \tag{8}$$

Thus, by defining $\pi_t = [Q_{x,1}^\theta(t)(\Delta t), \cdots, (1 + Q_{x,x}^\theta(t))\Delta t \cdots, Q_{x,N}^\theta(t)(\Delta t)]$, we sample $x_{t+\Delta t} \sim \text{Cat}(\pi_t)$, where $\text{Cat}(\cdot)$ denotes the categorical distribution.

However, this procedure is not differentiable with respect to $\theta$, which limits its applicability for optimization. To address this, we first recognize that sampling from the categorical distribution can be reduced to a Gumbel-max operation. Although this operation itself remains non-differentiable, we can modify it by replacing the max operation with a softmax, as shown in Line 7, which is also utilized in discrete VAE (Jang et al., 2016). This modification results in a new variable, $\bar{x}_t \sim [0,1]^N$, which maintains differentiability with respect to $\theta$. As the temperature $\tau$ approaches zero, $\bar{x}_t$ converges to a sample from the exact categorical distribution $\text{Cat}(\pi_t)$, effectively becoming $x_t$. Thus, we typically set the temperature to a low value to closely approximate the true distribution.

---

**Algorithm 1 DRAKES** (Direct Reward bAcKpropagation with gumbEl Softmax trick)

---

1: **Require**: Pretrained diffusion models $Q^{\theta_{\mathrm{pre}}} : \mathbb{R}^N \times \mathbb{R}^N \to \mathbb{R}$, reward $r : \mathcal{X} \to \mathbb{R}$, learning rate $\beta$, Batch size $B$, Iteration $S$, Time-step $\Delta t$, Temperature $\tau$, Regularization parameter $\alpha$

2: **for** $s \in [1, \cdots, S]$ **do**

3:    **for** $i \in [1, \cdots, B]$ **do**

4:       **for** $t \in [0, \Delta t, \cdots, T]$ **do**

5:          Set $[\pi(t)_1, \cdots, \pi(t)_N] \in \Delta(\mathcal{X})$ where $\pi(t)_y = \begin{cases} [\bar{x}_{t-1}^{(i)}]_y + \Delta t \sum_{x \in \mathcal{X}} [\bar{x}_{t-1}^{(i)}]_x Q_{y,x}^{\theta_s} \ (t > 0) \\ p_{\lim}(y) \ (t = 0) \end{cases}$

6:          Sample $k \in [1, \cdots, N]; G_k \sim \mathrm{Gumbel}(0, 1)$

7:          Set a differentiable counterpart of the sample at time $t$:

$$\bar{x}_t^{(i)} \leftarrow \left[ \frac{\exp((\pi(t)_1 + G_1)/\tau)}{\sum_y \exp((\pi(t)_y + G_y)/\tau)}, \cdots, \frac{\exp((\pi(t)_N + G_N)/\tau)}{\sum_y \exp(\pi(t)_y + G_y)/\tau)} \right]$$

8:       **end for**

9:    **end for**

10:   Set the loss:

$$g(\theta_s) = \frac{1}{B} \sum_{i=1}^B \left[ r(\bar{x}_T^{(i)}) - \frac{\alpha}{T} \sum_{t=1}^T \sum_{x \in \mathcal{X}} [\bar{x}_{t-1}^{(i)}]_x \sum_{\substack{y \in \mathcal{X} \\ y \neq x}} \left\{ -Q_{x,y}^{\theta_s}(t) + Q_{x,y}^{\theta_{\mathrm{pre}}}(t) + Q_{x,y}^{\theta_s}(t) \log \frac{Q_{x,y}^{\theta_s}(t)}{Q_{x,y}^{\theta_{\mathrm{pre}}}(t)} \right\} \right]$$

11:   Update a parameter: $\theta_{s+1} \leftarrow \theta_s + \beta \nabla_\theta g(\theta)|_{\theta=\theta_s}$

12: **end for**

13: **Output**: $\theta_{S+1}$

---

**Stage 2: Optimization (Step 10-11)** After approximately sampling from the distribution induced by $P^{\theta_s}$, we update the parameter $\theta_s$ by maximizing the empirical objective. Although $x_t$ itself may not have a valid gradient, $\bar{x}_t$ retains the gradient with respect to $\theta$. Therefore, we use the empirical approximation based on $\bar{x}_t$. We offer several remarks below, with details in Appendix E:

- **Validity of $\bar{x}_t$**: While $\bar{x}_t$ does not strictly belong to $\mathcal{X}$, this is practically acceptable since the generator $Q^\theta(t)$ is parameterized as a map $\mathbb{R}^N \times \mathbb{R}^N \to \mathbb{R}$.

- **SGD Variants**: Although Line 11 uses the standard SGD update, any off-the-shelf SGD algorithm, such as Adam (Kingma, 2014), can be applied in practice.

- **Soft Calculation with $\bar{x}_t$**: Transition probability $\pi(t)_y$ and the KL divergence term in $g(\theta)$ are modified to their soft counterparts by using $\bar{x}_t$ in place of $x_t$.

- **Straight-Through Gumbel Softmax**: Non-relaxed computations can be used in the forward pass (in Line 10). This is commonly known as straight-through Gumbel softmax estimator.

- **Truncated Backpropagration**: In practice, it is often more effective to backpropagate from intermediate time steps rather than starting from $t = 0$. In practice, we adopted this truncation approach, as in Clark et al. (2023).

- **Optimization Objective $g(\theta)$**: For the masked diffusion models (3) that we utilized, $g(\theta)$ can be further simplified to reduce computational complexity, as detailed in Appendix E.2.

## 5 THEORY OF **DRAKES**

In this section, we provide an overview of the proof for Theorem 1. Based on the insights gained from this proof, we reinterpret state-of-the-art classifier guidance for discrete diffusion models (Nisonoff et al., 2024) from a new perspective.

### 5.1 PROOF SKETCH OF THEOREM 1

We define the optimal value function $V_t : \mathcal{X} \to \mathbb{R}$ as follows:

$$\mathbb{E}_{x_{t:T} \sim P^{\theta^\star}} \left[ r(x_T) - \alpha \int_{s=t}^T \sum_{y \neq x_s} \left\{ Q_{x_s,y}^{\theta^\star}(s) - Q_{x_s,y}^{\theta_{\mathrm{pre}}}(s) + Q_{x_s,y}^{\theta^\star}(s) \log \frac{Q_{x_s,y}^{\theta^\star}(s)}{Q_{x_s,y}^{\theta_{\mathrm{pre}}}(s)} \right\} ds \mid x_t = x \right]. \quad (9)$$

This is the expected return starting from state $x$ at time $t$ following the optimal policy. Once the optimal value function is defined, the optimal generator can be expressed in terms of this value function using the Hamilton-Jacobi-Bellman (HJB) equation in CTMC, as shown below.

**Theorem 2** (Optimal generator). *For $x \neq y$ $(x, y \in \mathcal{X})$, the optimal CTMC generator $Q_{x,y}^{\theta^\star}(t)$ can be expressed by the pretrained CTMC generator $Q_{x,y}^{\theta_{\mathrm{pre}}}(t)$ and the value function 9:*

$$Q_{x,y}^{\theta^\star}(t) = Q_{x,y}^{\theta_{\mathrm{pre}}}(t) \exp(\{V_t(y) - V_t(x)\}/\alpha). \tag{10}$$

Next, consider an alternative expression for the soft value function, derived using the Kolmogorov backward equations in CTMC. This expression is particularly useful for learning value functions.

**Theorem 3** (Feynman–Kac Formula in CTMC). *The value function 9 follows the Feynman–Kac formulation with respect to the pretrained CTMC distribution $P^{\theta_{\mathrm{pre}}}$ generated by $Q^{\theta_{\mathrm{pre}}}$ as follows*

$$\exp(V_t(x)/\alpha) = \mathbb{E}_{x_{t:T} \sim P^{\theta_{\mathrm{pre}}}}[\exp(r(x_T)/\alpha)|x_t = x] \tag{11}$$

With this preparation, we can prove our main theorem, which reduces to Theorem 1 when $t = T$.

**Theorem 4** (Marginal distribution induced by the optimal generator $Q^{\theta^\star}(t)$). *The marginal distribution at time $t$ by the CTMC trajectory in (6) with generator $Q^{\theta^\star}(t)$, i.e., $p_t^\star \in \Delta(\mathcal{X})$, satisfies*

$$p_t^\star(\cdot) \propto \exp(V_t(\cdot)/\alpha) p_t^{\mathrm{pre}}(\cdot), \tag{12}$$

*where $p_t^{\mathrm{pre}} \in \Delta(\mathcal{X})$ is a marginal distribution induced by pretrained CTMC at $t$.*

This is proved by showing the Kolmogorov forward equation in CTMC: $dp_t^\star/dt = Q^{\theta^\star}(t)p_t^\star$.

## 5.2 RELATION TO CLASSIFIER GUIDANCE FOR DISCRETE DIFFUSION MODELS

Now, we derive an alternative fine-tuning-free algorithm by leveraging observations in Section 5.1 for reward maximization. If we can directly obtain the optimal generator $Q^{\theta^\star}$, we can achieve our objective. Theorem 2 suggests that the optimal generator $Q^{\theta^\star}$ is a product of the generator from the pretrained model and the value functions. Although we don't know the exact value functions, they can be learned through regression using Theorem 3 based on

$$\exp(V_t(\cdot)/\alpha) = \underset{g:\mathcal{X}\to\mathbb{R}}{\operatorname{argmin}} \mathbb{E}_{x_T \sim P^{\theta_{\mathrm{pre}}}(x_T|x_t)}[\{\exp(r(x_T)/\alpha) - g(x_t)\}^2]. \tag{13}$$

In practice, while we can't calculate the exact expectation, we can still replace it with its empirical analog. Alternatively, we can approximate it by using a map from $x_t$ to $x_0$ in pretrained models following DPS (Chung et al., 2022b) or reconstruction guidance (Ho et al., 2022).

Interestingly, a similar algorithm was previously proposed by Nisonoff et al. (2024). While Nisonoff et al. (2024) originally focused on conditional generation, their approach can also be applied to reward maximization or vice versa. In their framework for conditional generation, they define $r(x) = \log p(z|x)$ (e.g., the log-likelihood from a classifier) and set $\alpha = 1$. By adapting Theorem 2 and 3 to their setting, we obtain:

$$Q_{x,y}^{\theta^\star}(t) = Q_{x,y}^{\theta_{\mathrm{pre}}}(t) \times p_t(z|y)/p_t(z|x), \quad p_t(z|x_t) := \mathbb{E}_{x_{t:T} \sim P^{\theta_{\mathrm{pre}}}}[p(z|x_T) \mid x_t]. \tag{14}$$

Thus, we can rederive the formula in Nisonoff et al. (2024). Here, we also note that this type of result is commonly referred to as the Doob transform in the literature on stochastic processes (Levin & Peres, 2017, Section 17).

While this argument suggests that classifier guidance and RL-based fine-tuning approaches theoretically achieve the same goal in an ideal setting (without function approximation, sampling, or optimization errors), their practical behavior can differ significantly, as we demonstrate in Section 6. At a high level, the advantage of classifier guidance is that it requires no fine-tuning, but the inference time may be significantly longer due to the need to recalculate the generator during inference. Indeed, this classifier guidance requires $O(NM)$ computations of value functions at each step to calculate the normalizing constant. While this can be mitigated using a Taylor approximation, there is no theoretical guarantee for this heuristic in discrete diffusion models. Lastly, learning value functions in classifier guidance can often be practically challenging.

## 6 EXPERIMENTS

Our experiments focus on the design of regulatory DNA sequences for enhancer activity and protein sequences for stability. Our results include comprehensive evaluations, highlighting the ability of **DRAKES** to produce natural-like sequences while effectively optimizing the desired properties. We also experiment in text generation for minimizing toxicity, detailed in Appendix F.4.

### 6.1 BASELINES

We compare **DRAKES** against several baseline methods discussed in Section 2, which we summarize below with further details in Appendix F.1.

- **Guidance-based Methods (CG, SMC, TDS).** We compare our approach with representative guidance-based methods, including state-of-the-art classifier guidance (**CG**) tailored to discrete diffusion models (Nisonoff et al., 2024), SMC-based guidance methods (e.g., Wu et al. (2024)): **SMC**, where the proposal is a pretrained model and **TDS**, where the proposal is CG.
- **Classifier-free Guidance (CFG)** (Ho & Salimans, 2022). CFG is trained on labeled datasets with the measured attributes we aim to optimize.
- **Pretrained.** We generated sequences using pretrained models without fine-tuning.
- **DRAKES w/o KL.** This ablation of **DRAKES** omits the KL regularization term, evaluating how well this term mitigates over-optimization (discussed in Section 4.1).

### 6.2 REGULATORY DNA SEQUENCE DESIGN

Here we aim to optimize the activity of regulatory DNA sequences such that they drive gene expression in specific cell types, a critical task for cell and gene therapy (Taskiran et al., 2024).

**Dataset and settings.** We experiment on a publicly available large-scale enhancer dataset (Gosai et al., 2023), which measures the enhancer activity of $\sim$700k DNA sequences (200-bp length) in human cell lines using massively parallel reporter assays (MPRAs), where the expression driven by each sequence is measured. We pretrain the masked discrete diffusion model (Sahoo et al., 2024) on all the sequences. We then split the dataset and train two reward oracles (one for finetuning and one for evaluation) on each subset, using the Enformer (Avsec et al., 2021) architecture to predict the activity level in the HepG2 cell line. These datasets and reward models are widely used in the literature on computational enhancer design (Lal et al., 2024; Uehara et al., 2024c; Sarkar et al., 2024). Detailed information about the pretrained model and reward oracles is in Appendix F.2.

**Evaluations.** To comprehensively evaluate each model's performance in enhancer generation, we use the following metrics:

- *Predicted activity based on the evaluation reward oracle (Pred-Activity).* We predict the enhancer activity level in the HepG2 cell line using the reward oracle trained on the evaluation subset. Note that the models are fine-tuned (or guided) with the oracle trained on a *different* subset of the data, splitting based on chromosome following conventions (but in the same cell lines) (Lal et al., 2024).
- *Binary classification on chromatin accessibility (ATAC-Acc).* We use an independent binary classification model trained on chromatin accessibility data in the HepG2 cell line (Consortium et al., 2012) (active enhancers should have accessible chromatin). While this is *not* used for fine-tuning, we use it for evaluation to further validate the predicted activity of the synthetic sequences, following Lal et al. (2024).
- *3-mer Pearson correlation (3-mer Corr).* We calculate the 3-mer Pearson correlation between the synthetic sequences and the sequences in the dataset (Gosai et al., 2023) with top 0.1% HepG2 activity level. Models that generate sequences that are more natural-like and in-distribution have a higher correlation.
- *JASPAR motif analysis (JASPAR Corr).* We scan the generated sequences of each model with JASPAR transcription factor binding profiles Castro-Mondragon et al. (2022), which identify potential transcription factor binding motifs in the enhancer sequences (which are expected to drive enhancer activity). We then count the occurrence frequency of each motif and calculate the Spearman correlation of motif frequency between the synthetic sequences generated by each model and the top 0.1% HepG2 activity sequences in the dataset.
- *Approximated log-likelihood of sequences (App-Log-Lik).* We calculate the log-likelihood of the generated sequences with respect to the pretrained model to measure how likely the sequences

are to be natural-like. Models that over-optimize the reward oracle generate out-of-distribution sequences and would have a low likelihood to the pretrained model. The likelihood is calculated using the ELBO of the discrete diffusion model in Sahoo et al. (2024).

**Results.** **DRAKES** generates sequences with high predicted activity in the HepG2 cell line, as robustly measured by *Pred-Activity* and *ATAC-Acc* (Table 1). The generated sequences closely resemble natural enhancers, as indicated by high 3-mer and JASPAR motif correlations, and a similar likelihood to the pretrained model. These highlight **DRAKES**'s effectiveness in generating plausible high-activity enhancer sequences. Notably, while **DRAKES**, without KL regularization achieves higher *Pred-Activity*, this can be attributed to over-optimization. Despite splitting the data for fine-tuning and evaluation, the sequences remain highly similar due to many analogous regions within each chromosome. However, when evaluated with an independent activity oracle, *ATAC-Acc*, **DRAKES** demonstrates superior performance while maintaining higher correlations and log-likelihood. Ablations on gradient truncation and Gumbel Softmax schedule are in Appendix F.2.

Table 1: Model performance on regulatory DNA sequence design. **DRAKES** generates sequences with high activity in the HepG2 cell line, measured by *Pred-Activity* and *ATAC-Acc*, while being natural-like by high 3-mer and JASPAR motif correlations and likelihood. We report the mean across 3 random seeds, with standard deviations in parentheses.

| Method | Pred-Activity (median)↑ | ATAC-Acc↑ (%) | 3-mer Corr↑ | JASPAR Corr↑ | App-Log-Lik (median)↑ |
|---|---|---|---|---|---|
| Pretrained | 0.17(0.04) | 1.5(0.2) | -0.061(0.034) | 0.249(0.015) | -261(0.6) |
| CG | 3.30(0.00) | 0.0(0.0) | -0.065(0.001) | 0.212(0.035) | -266(0.6) |
| SMC | 4.15(0.33) | 39.9(8.7) | 0.840(0.045) | 0.756(0.068) | -259(2.5) |
| TDS | 4.64(0.21) | 45.3(16.4) | 0.848(0.008) | 0.846(0.044) | **-257(1.5)** |
| CFG | 5.04(0.06) | 92.1(0.9) | 0.746(0.001) | 0.864(0.011) | -265(0.6) |
| **DRAKES** w/o KL | **6.44(0.04)** | 82.5(2.8) | 0.307(0.001) | 0.557(0.015) | -281(0.6) |
| **DRAKES** | 5.61(0.07) | **92.5(0.6)** | **0.887(0.002)** | **0.911(0.002)** | -264(0.6) |

## 6.3 PROTEIN SEQUENCE DESIGN: OPTIMIZING STABILITY IN INVERSE FOLDING MODEL

In this task, given a pretrained inverse folding model that generates sequences conditioned on the backbone's conformation (3D structure), our goal is to optimize the stability of these generated sequences, following Widatalla et al. (2024).

**Dataset and settings.** First, we pretrained an inverse folding model based on the diffusion model (Campbell et al., 2024) and the ProteinMPNN (Dauparas et al., 2022) architecture, using the PDB training set from Dauparas et al. (2022). Next, we trained the reward oracles using a different large-scale protein stability dataset, Megascale (Tsuboyama et al., 2023), which includes stability measurements (i.e., the Gibbs free energy change) for ∼1.8M sequence variants from 983 natural and designed domains. Following dataset curation and a train-validation-test splitting procedure from Widatalla et al. (2024) (which leads to ∼0.5M sequences on 333 domains) and using the ProteinMPNN architecture, we constructed two reward oracles – one for fine-tuning and one for evaluation, that predict stability from the protein sequence and wild-type conformation. Detailed information on the pretrained model and reward oracles is in Appendix F.3.

**Evaluations.** We use the following metrics to evaluate the stability of the generated sequences and their ability to fold into the desired structure. During evaluation, we always condition on protein backbone conformations from the test data that are *not* used during fine-tuning.

- *Predicted stability on the evaluation reward oracle (Pred-ddG).* The evaluation oracle is trained with the full Megascale dataset (train+val+test) to predict protein stability. Conversely, the fine-tuning oracle is trained only with data from the Megascale training set. Thus, during fine-tuning, the algorithms do not encounter any proteins used for evaluation.
- *Self-consistency RMSD of structures (scRMSD).* To assess how well a generated sequence folds into the desired structure, we use ESMFold (Lin et al., 2023) to predict the structures of the generated sequences and calculate their RMSD relative to the wild-type structure (i.e., the original backbone structure we are conditioning on). This is a widely used metric (Campbell et al., 2024; Trippe et al., 2022; Chu et al., 2024).

Following prior works (Campbell et al., 2024; Nisonoff et al., 2024), we calculate the success rate of inverse folding as the ratio of generated sequences with *Pred-ddG*> 0 and *scRMSD*< 2.

Table 2: Model performance on inverse protein folding. **DRAKES** generates protein sequences that have high stability and fold to the desired structure, outperforming baselines in the overall success rate. We report the mean across 3 random seeds, with standard deviations in parentheses.

| Method | Pred-ddG (median)↑ | %(ddG> 0) (%)↑ | scRMSD (median)↓ | %(scRMSD< 2)(%)↑ | Success Rate (%)↑ |
|---|---|---|---|---|---|
| Pretrained | -0.544(0.037) | 36.6(1.0) | 0.849(0.013) | 90.9(0.6) | 34.4(0.5) |
| CG | -0.561(0.045) | 36.9(1.1) | 0.839(0.012) | 90.9(0.6) | 34.7(0.9) |
| SMC | 0.659(0.044) | 68.5(3.1) | 0.841(0.006) | 93.8(0.4) | 63.6(4.0) |
| TDS | 0.674(0.086) | 68.2(2.4) | **0.834(0.001)** | **94.4(1.2)** | 62.9(2.8) |
| CFG | -1.186(0.035) | 11.0(0.4) | 3.146(0.062) | 29.4(1.0) | 1.3(0.4) |
| **DRAKES** w/o KL | **1.108(0.004)** | **100.0(0.0)** | 7.307(0.054) | 34.1(0.2) | 34.1(0.2) |
| **DRAKES** | 1.095(0.026) | 86.4(0.2) | 0.918(0.006) | 91.8(0.5) | **78.6(0.7)** |

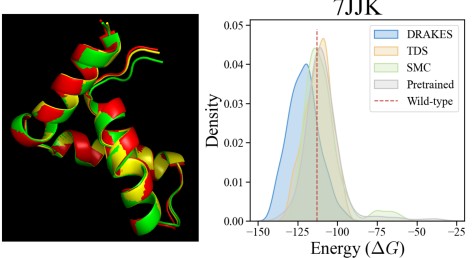

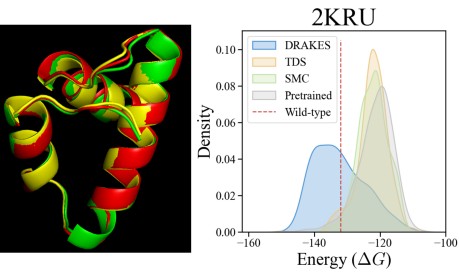

(a) Conditioning on the backbone structure of 7JJK.     (b) Conditioning on the backbone structure of 2KRU.

Figure 2: Examples of generated proteins. Red: Wild-type backbone structure (the one we condition on), Yellow: Structure predicted by ESMFold from the wild-type (true) sequence, Green: Structure predicted by ESMFold from the sequence generated by **DRAKES**. The structures for sequences generated by **DRAKES** show good alignment with the original structure (the scRMSDs are $0.768$ for 7JJK and $0.492$ for 2KRU). Histograms: Gibbs free energy for each generated sequence, calculated using physics-based simulations. In these two cases, the sequences generated by **DRAKES** appear to be more stable than the baselines.

**Results.** For inverse protein folding, **DRAKES** generates high-stability protein sequences capable of folding into the conditioned structure (Table 2). It achieves the highest *Pred-ddG* among all methods, while maintaining a similar success rate of inverse folding (measured by %(scRMSD< 2), the percentage of *scRMSD* smaller than 2) as the pretrained model. Considering both factors, **DRAKES** significantly outperforms all baseline methods in terms of overall success rate. Note that CFG does not work well for protein sequence design due to limited labeled data, as Megascale includes only a few hundred backbones, making generalization difficult. This is expected, as we mention in Section 2. Further details are provided in Appendix F.1. We provide additional results with pLDDT and sequence diversity, as well as ablation studies on the gradient truncation number and Gumbel Softmax temperature schedule in Appendix F.3.

Moreover, the results highlight the importance of the KL term, as **DRAKES** without KL regularization tends to suffer from over-optimization, with high *scRMSD* (i.e., failing to fold back to the target backbone structure), even though *Pred-ddG* may remain high.

**In silico validation.** For validation purposes, we calculate the stability (i.e., Gibbs free energy) of the generated sequences using physics-based simulations (PyRosetta (Chaudhury et al., 2010)) for wild-type protein backbone structures in Figure 2, following (Widatalla et al., 2024). Although all models are conditioned on the same set of protein backbones, different sets of sequences generated by generative methods can lead to significant differences in side chain interactions, which affect folding energies. The results demonstrate that sequences generated by **DRAKES** are more stable in this validation compared to other baselines. For additional results, refer to Figure 6 in Appendix F.3.

## 7 CONCLUSIONS

We propose a novel algorithm that incorporates reward maximization into discrete diffusion models, leveraging the Gumbel-Softmax trick to enable differentiable reward backpropagation, and demonstrate its effectiveness in generating DNA and protein sequences optimized for task-specific objectives. For future work, we plan to conduct more extensive in silico validation and pursue wet-lab validation.

## REPRODUCIBILITY STATEMENT

For the theoretical results presented in the paper, we provide explanations of assumptions and complete proofs in Appendix C. For the proposed algorithm and experimental results, we provide detailed explanations of the algorithm implementations and experimental setup in Section 4.2, Section 6, Appendix E, and Appendix F, and the code and data can be found in `https://github.com/ChenyuWang-Monica/DRAKES`. For the datasets used in the experiments, we utilize publicly available datasets and elaborate the data processing procedures in Section 6 and Appendix F.

## ACKNOWLEDGEMENT

CW and TJ acknowledge support from the Machine Learning for Pharmaceutical Discovery and Synthesis (MLPDS) consortium, the DTRA Discovery of Medical Countermeasures Against New and Emerging (DOMANE) threats program, and the NSF Expeditions grant (award 1918839) Understanding the World Through Code.

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

## A  More Related Works

**Dirichlet diffusion models for discrete spaces.**  Another approach to diffusion models for discrete spaces has been proposed (Stark et al., 2024; Avdeyev et al., 2023; Zhou et al., 2024). In these models, each intermediate state is represented as a vector within a simplex. This is in contrast to masked diffusion models, where each state is a discrete variable.

**Relative trajectory balance for posterior inference.** Venkatraman et al. (2024) proposed to use relative trajectory balance to train a diffusion model that sample from a posterior $x \sim p^{\text{post}}(x) \propto p(x)r(x)$. When $r(x)$ is defined as the exponential reward, it can be utilized for reward optimization. However, this approach requires estimation of the normalizing constant term of the unnormalized density. In contrast, we solve a control problem with direct backpropagation. Besides, Venkatraman et al. (2024) models continuous diffusion as a Markovian generative process and is not specifically tailored to discrete diffusion models.

## B  Potential Limitations

We have formulated the RL problem, (5), in the context of CTMC. The proposed algorithm in our paper to solve this problem requires reward models to be differentiable. Since differentiable models are necessary when working with experimental offline data, this assumption is not overly restrictive. Moreover, many state-of-the-art sequence models mapping sequences to functions in biology are available today, such as enformer borozi. In cases where creating differentiable models is challenging, we recommend using PPO-based algorithms (Black et al., 2023; Fan et al., 2023) or reward-weighted MLE (Uehara et al., 2024a, Section 6.1).

## C  Proof of Theorems

### C.1  Preparation

We prepare two well-known theorems in CTMC for the pedagogic purpose. For example, refer to Yin & Zhang (2012) for the more detailed proof. In these theorems, we suppose we have the CTMC:

$$\frac{dp_t}{dt} = Q(t)p_t. \tag{15}$$

**Lemma 1** (Kolmogorov backward equation). *We consider $g(\cdot, t) = \mathbb{E}[r(x_T)|x_t = \cdot]$ where the expectation is taken w.r.t.* (15). *Then, this function $g : \mathcal{X} \times [0, T] \to \mathbb{R}$ is characterized by the following ODE:*

$$\frac{dg(x, t)}{dt} = \sum_{y \neq x} Q_{x,y}(t)\{g(x, t) - g(y, t)\}, \quad g(x, T) = r(x_T).$$

*Proof.* Here, we prove that the p.d.f. $g$ satisfies the above backward equation. To show the converse, we technically require regularity conditions to claim the ODE solution is unique, which can often be proved by the contraction mapping theorem under certain regularity conditions. Since this is a well-known result, we skip the converse part. If readers are interested in details, we refer to Yin & Zhang (2012).

When $t = T$, the statement $g(x, T) = r(x_T)$ is obvious. For the rest of the proof, we aim to show a result when $t \neq T$. We have

$$g(x_t, t) = \int g(x_{t+dt}, t + dt)p(x_{t+dt}|x_t)dx_{t+dt}.$$

The above implies

$$g(x, t) = \{1 + Q_{x,x}(t)dt\}g(x, t + dt) + \sum_{y \neq x}\{Q_{x,y}(t)dt\}g(y, t + dt).$$

Now combined with the property of the generator as follows

$$0 = \sum_y Q_{x,y}(t+dt),$$

With some algebra,

$$g(x,t) = g(x,t+dt) - \sum_{y\neq x}\{Q_{x,y}(t)dt\}g(x,t+dt) + \sum_{y\neq x}\{Q_{x,y}(t)dt\}g(y,t+dt).$$

Then, we have

$$\frac{g(x,t) - g(x,t+dt)}{dt} = \sum_{y\neq x}Q_{x,y}(t)\{g(y,t+dt) - g(x,t+dt)\}$$

Finally, by setting $dt \to 0$, we obtain

$$-\frac{dg(x,t)}{dt} = \sum_{y\neq x}Q_{x,y}(t)\{g(y,t) - g(x,t)\}.$$

$\square$

**Lemma 2** (Kolmogorov forward equation). *The density $p_t \in \Delta(\mathcal{X})$ is characterized as the following ODE:*

$$\frac{dp_t(x)}{dt} = \sum_{y\neq x}Q_{y,x}(t)p_t(y) - \sum_{y\neq x}Q_{x,y}(t)p_t(x), \quad p_0 = p_{\mathrm{ini}}.$$

*Proof.* Here, we prove that the p.d.f. $p_t$ satisfies the above forward equation. To show the converse, we technically require regularity conditions to claim the ODE solution is unique, which can often be proved by the contraction mapping theorem. Here, we skip the converse part.

We first have

$$p_{t+dt}(x) = \int p_{t+dt}(x|x_t)p_t(x_t)dx_t$$

This implies

$$p_{t+dt}(x) = \sum_{y\neq x}\{Q_{y,x}(t)dt\,p_t(y)\} + \{1 + Q_{x,x}(t)dt\}p_t(x)$$

$$= \sum_{y\neq x}\{Q_{y,x}(t)dt\,p_t(y)\} + \{1 - \sum_{y\neq x}Q_{x,y}(t)dt\}p_t(x).$$

Hence,

$$\frac{p_{t+dt}(x) - p_t(x)}{dt} = \sum_{y\neq x}Q_{y,x}(t)p_t(y) - \sum_{y\neq x}Q_{x,y}(t)p_t(x).$$

By taking $dt \to 0$, we obtain

$$\frac{dp_t(x)}{dt} = \sum_{y\neq x}Q_{y,x}(t)p_t(y) - \sum_{y\neq x}Q_{x,y}(t)p_t(x).$$

Then, the proof is completed. $\square$

### C.2 Proof of Theorem 2

We derive the Hamilton-Jacobi-Bellman (HJB) equation in CTMC. For this purpose, we consider the recursive equation:

$$V(x,t) = \max_\theta \left[ \left\{ \sum_{y\neq x} Q_{x,y}^\theta(t) - Q_{x,y}^{\theta_{\mathrm{pre}}}(t) - Q_{x,y}^\theta(t)\log\frac{Q_{x,y}^\theta(t)}{Q_{x,y}^{\theta_{\mathrm{pre}}}(t)} \right\} dt \right.$$

$$\left. + \sum_{y\neq x}\{Q_{x,y}^\theta(t)dt\,V(y,t+dt)\} + \{1 + Q_{x,x}^\theta(t)\}V(x,t+dt)\} \right].$$

Using $\sum_{y \in \mathcal{X}} Q_{x,y}(t) = 0$, this is equal to

$$V(x,t) = \max_{\theta} \left[ \left\{ \sum_{y \neq x} Q_{x,y}^{\theta}(t) - Q_{x,y}^{\theta_{\text{pre}}}(t) - Q_{x,y}^{\theta}(t) \log \frac{Q_{x,y}^{\theta}(t)}{Q_{x,y}^{\theta_{\text{pre}}}(t)} \right\} dt \right.$$

$$\left. + V(x, t+dt) + \sum_{y \neq x} Q_{x,y}^{\theta}(t) dt \{ V(y, t+dt) - V(x, t+dt) \} \right].$$

By taking $dt$ to 0, the above is equal to

$$-\frac{dV(x,t)}{dt} = \max_{\theta \in \Theta} \left\{ \left[ \sum_{y \neq x} Q_{x,y}^{\theta}(t) - Q_{x,y}^{\theta_{\text{pre}}}(t) - Q_{x,y}^{\theta}(t) \log \frac{Q_{x,y}^{\theta}(t)}{Q_{x,y}^{\theta_{\text{pre}}}(t)} \right] + \sum_{y \neq x} Q_{x,y}^{\theta}(t) \{ V(y,t) - V(x,t) \} \right\}$$

(16)

This is the HJB equation in CTMC.

Finally, with simple algebra (i.e., taking functional derivative under the constraint $0 = \sum_{y \in \mathcal{X}} Q_{x,y}^{\theta}(t)$), we can show

$$\forall x \neq y; Q_{x,y}^{\theta^{\star}}(t) = Q_{x,y}^{\theta_{\text{pre}}}(t) \exp(\{ V(y,t) - V(x,t) \}).$$

### C.3    PROOF OF THEOREM 3

This theorem is proved by invoking the Kolmogorov backward equation.

First, by plugging

$$\forall x \neq y; Q_{x,y}^{\theta^{\star}}(t) = Q_{x,y}^{\theta_{\text{pre}}}(t) \exp(\{ V(y,t) - V(x,t) \}).$$

into (16), we get

$$\frac{dV(x,t)}{dt} = \sum_{y \neq x} Q_{x,y}^{\theta_{\text{pre}}}(t) \{ 1 - \exp(\{ V(y,t) - V(x,t) \}) \}.$$

By multiplying $\exp(V(x,t))$ to both sides, it reduces to

$$\frac{d \exp(V(x,t))}{dt} = \sum_{y \neq x} Q_{x,y}^{\theta_{\text{pre}}}(t) \{ \exp(V(x,t)) - \exp(V(y,t)) \}.$$

(17)

Furthermore, clearly, $V(x,T) = r(x_T)$. Then, the statement is proved by invoking the Kolmogorov backward equation.

### C.4    PROOF OF THEOREM 4

We define

$$H_t(x) := \exp(V(x,t)) p_t(x)/C.$$

We aim to prove that the above satisfies the Kolmogorov forward equation:

$$\underbrace{\frac{dH_t(x)}{dt}}_{\text{l.h.s.}} = \underbrace{\sum_{y \neq x} Q_{y,x}^{\theta^{\star}}(t) H_t(y) - \sum_{y \neq x} Q_{x,y}^{\theta^{\star}}(t) H_t(x)}_{\text{r.h.s.}}, \quad p_{\text{ini}} = H_0(\cdot).$$

**Remark 4.** *Here, note that $p_{\text{ini}} = \delta(\cdot = \text{Mask}) = H_0(\cdot)$ holds because we have considered a scenario where the initial is the Dirac delta distribution. When the original initial distribution is stochastic, we will see in Appendix D that we still have $p_{\text{ini}} = H_0(\cdot)$.*

First, we calculate the l.h.s. Here, recall

$$\frac{d \exp(V(x,t))}{dt} = \sum_{y \neq x} Q_{x,y}^{\theta_{\mathrm{pre}}}(t)\{\exp(V(x,t)) - \exp(V(y,t))\}$$

using (17), and

$$\frac{dp_t(x)}{dt} = \sum_{y \neq x} Q_{y,x}^{\theta_{\mathrm{pre}}}(t)p_t(y) - \sum_{y \neq x} Q_{x,y}^{\theta_{\mathrm{pre}}}(t)p_t(x)$$

holds, using the Kolmogorov forward equation. Then, we obtain

$$\frac{dH_t(x)}{dt} = \frac{1}{C} \times \left\{ \frac{d \exp(V(x,t))}{dt} p_t(x) + \exp(V(x,t)) \frac{dp_t(x)}{dt} \right\}$$

$$= \frac{1}{C} \times \left[ \sum_{y \neq x} Q_{x,y}^{\theta_{\mathrm{pre}}}(t)\{\exp(V(x,t)) - \exp(V(y,t))\}p_t(x) \right]$$

$$+ \frac{1}{C} \times \exp(V(x,t)) \left\{ \sum_{y \neq x} Q_{y,x}^{\theta_{\mathrm{pre}}}(t)p_t(y) - \sum_{y \neq x} Q_{x,y}^{\theta_{\mathrm{pre}}}(t)p_t(x) \right\}$$

$$= \frac{1}{C} \times \sum_{y \neq x} Q_{y,x}^{\theta_{\mathrm{pre}}}(t) \exp(V(x,t))p_t(y) - \frac{1}{C} \times \sum_{y \neq x} Q_{x,y}^{\theta_{\mathrm{pre}}}(t) \exp(V(y,t))p_t(x).$$

On the other hand, the r.h.s. is

$$\frac{1}{C} \times \left\{ \sum_{y \neq x} Q_{y,x}^{\theta^\star}(t)H_t(y) - \sum_{y \neq x} Q_{x,y}^{\theta^\star}(t)H_t(x) \right\}$$

$$= \frac{1}{C} \times \sum_{y \neq x} Q_{y,x}^{\theta_{\mathrm{pre}}}(t) \exp(\{V(x,t) - V(y,t)\})H_t(y) - \frac{1}{C} \sum_{y \neq x} Q_{x,y}^{\theta_{\mathrm{pre}}}(t) \exp(\{V(y,t) - V(x,t)\})H_t(x)$$

$$= \frac{1}{C} \times \sum_{y \neq x} Q_{y,x}^{\theta_{\mathrm{pre}}}(t) \exp(V(x,t))p_t(y) - \frac{1}{C} \times \sum_{y \neq x} Q_{x,y}^{\theta_{\mathrm{pre}}}(t) \exp(V(y,t))p_t(x).$$

Here, from the first line to the second line, we use

$$\forall x \neq y; Q_{x,y}^{\theta^\star}(t) = Q_{x,y}^{\theta_{\mathrm{pre}}}(t) \exp(\{V(y,t) - V(x,t)\}).$$

Finally, we can see that $l.h.s. = r.h.s.$ Furthermore, recalling we have an assumption that $p_{\mathrm{ini}}$ is Dirac delta distribution, we clearly have $p_{\mathrm{ini}} = H_0(\cdot)$. Hence, the statement is proved by the Kolmogorov forward equation.

## D    EXTENSION WITH STOCHASTIC INITIAL DISTRIBUTIONS

When initial distributions are stochastic, we need to modify algorithms. Although there are several strategies to take stochastic initial distributions into account (Uehara et al., 2024b; Domingo-Enrich et al., 2024), we consider the strategy of learning the initial distributions (Uehara et al., 2024b).

Hence, we consider the following control problem:

$$\theta^\star = \underset{\theta}{\mathrm{argmax}} \underbrace{\mathbb{E}_{x_{0:T} \sim P^\theta, x_0 \sim P_0^\theta} [r(x_T)]}_{\text{Reward term}} \tag{18}$$

$$- \alpha \underbrace{\left\{ \mathbb{E}_{x_{0:T} \sim P^\theta, x_0 \sim P_0^\theta} \left[ \int_{t=0}^{T} \sum_{y \neq x_t} \left\{ Q_{x_t,y}^{\theta_{\mathrm{pre}}}(t) - Q_{x_t,y}^{\theta}(t) + Q_{x_t,y}^{\theta}(t) \log \frac{Q_{x_t,y}^{\theta}(t)}{Q_{x_t,y}^{\theta_{\mathrm{pre}}}(t)} \right\} dt \right] + \mathrm{KL}(P_0^\theta \mid p_{\mathrm{lim}}) \right\}}_{\text{KL term}}.$$

Compared to the control problem (5), in our new formulation (18), we parameterize not only generators but also initial distributions. In this case, Theorem 1 still holds.

### D.1 PROOF

Recalling the definition of value functions:

$$V(k,t) = \mathbb{E}_{x_{k:T}\sim P^\theta}\left[r(x_T) - \alpha\int_{t=k}^T \sum_{y\neq x_t}\left\{Q_{x_t,y}^{\theta_{\mathrm{pre}}}(t) - Q_{x_t,y}^\theta(t) + Q_{x_t,y}^\theta(t)\log\frac{Q_{x_t,y}^\theta(t)}{Q_{x_t,y}^{\theta_{\mathrm{pre}}}(t)}\right\}dt\right],$$

the optimal initial distribution needs to satisfy

$$\operatorname*{argmax}_\theta \mathbb{E}_{x_0\sim P_0^\theta}[V_0(x_0)] - \alpha\mathrm{KL}(P_0^\theta\mid p_{\mathrm{lim}}).$$

This is equal to the distribution proportional to

$$\exp(V_0(x_0)/\alpha)p_{\mathrm{lim}}(x_0).$$

The rest of the proof holds as in the proof of Section C.4 without any change, referring to Remark 4.

## E DETAILS OF ALGORITHM

### E.1 STRAIGHT-THROUGH GUMBEL SOFTMAX

We apply the straight-through Gumbel softmax estimator to the last time step, i.e.

$$\mathrm{ST}(x_T^{(i)}) := \bar{x}_T^{(i)} + \mathrm{SG}(x_T^{(i)} - \bar{x}_T^{(i)})$$

where $x_T^{(i)}$ is the corresponding Gumbel-max variable, i.e. $x_T^{(i)} = \operatorname{argmax}_{x\in\mathcal{X}}[\bar{x}_T^{(i)}]_x$, and SG denotes stop gradient. Then, $\mathrm{ST}(x_T^{(i)})$ is input into the reward function $r(.)$ instead of $\bar{x}_T^{(i)}$ for forward and backward propagation.

We observe a boost in fine-tuning performance with the straight-through Gumbel softmax, as converting the input to $r(.)$ into a one-hot vector makes it better aligned with the reward oracle's training distribution.

### E.2 SIMPLIFIED FORMULA OF $g(\theta)$

The key objective function in **DRAKES**, $g(\theta)$, can be further simplified for the masked diffusion models that we utilized in the experiments.

$$g(\theta) = \frac{1}{B}\sum_{i=1}^B\left[r(\bar{x}_T^{(i)}) - \frac{\alpha}{T}\sum_{t=1}^T\sum_{x\in\mathcal{X}}[\bar{x}_{t-1}^{(i)}]_x\sum_{\substack{y\in\mathcal{X}\\y\neq x}}\left\{-Q_{x,y}^\theta(t) + Q_{x,y}^{\theta_{\mathrm{pre}}}(t) + Q_{x,y}^\theta(t)\log\frac{Q_{x,y}^\theta(t)}{Q_{x,y}^{\theta_{\mathrm{pre}}}(t)}\right\}\right]$$

We denote the second term estimating the KL divergence with the $i$-th sample as $k_i(\theta)$:

$$k^{(i)}(\theta) = \frac{1}{T}\sum_{t=1}^T\sum_{x\in\mathcal{X}}[\bar{x}_{t-1}^{(i)}]_x\sum_{\substack{y\in\mathcal{X}\\y\neq x}}\left\{-Q_{x,y}^\theta(t) + Q_{x,y}^{\theta_{\mathrm{pre}}}(t) + Q_{x,y}^\theta(t)\log\frac{Q_{x,y}^\theta(t)}{Q_{x,y}^{\theta_{\mathrm{pre}}}(t)}\right\}$$

When $x = \mathrm{Mask}$, the value of $Q_{x,y}(t)$ is irrelevant to the parametrization $\theta$, i.e.

$$Q_{x,y}^\theta(t) = Q_{x,y}^{\theta_{\mathrm{pre}}}(t) = \begin{cases}0, y\neq\mathrm{Mask}\\-\gamma, y = \mathrm{Mask}\end{cases}$$

where $\gamma$ is a constant related to the forward process schedule (Sahoo et al., 2024). In particular, when applying a linear schedule (as in our experiments), $\gamma = 1/t$. Thus, the corresponding KL divergence component equals 0.

When $x \neq \text{Mask}$,

$$Q_{x,y}^{\theta}(t) = \begin{cases} 0, y \neq \text{Mask} \\ \gamma \mathbb{E}_{\theta}[x_0 = x | x_{t-1} = \text{Mask}], y = \text{Mask} \end{cases}$$

Denote $\mathbb{E}_{\theta}[x_0 = x | x_{t-1} = \text{Mask}]$ as $[\hat{x}_0^{\theta}]_x$. The KL divergence term $k^{(i)}(\theta)$ can be simplified as

$$k^{(i)}(\theta) = \frac{1}{T} \sum_{t=1}^{T} \sum_{x \in \mathcal{X}} [\bar{x}_{t-1}^{(i)}]_x \sum_{\substack{y \in \mathcal{X} \\ y \neq x}} \left\{ -Q_{x,y}^{\theta}(t) + Q_{x,y}^{\theta_{\text{pre}}}(t) + Q_{x,y}^{\theta}(t) \log \frac{Q_{x,y}^{\theta}(t)}{Q_{x,y}^{\theta_{\text{pre}}}(t)} \right\}$$

$$= \frac{1}{T} \sum_{t=1}^{T} \sum_{x \in \mathcal{X}} [\bar{x}_{t-1}^{(i)}]_x \left\{ -Q_{x,\text{Mask}}^{\theta}(t) + Q_{x,\text{Mask}}^{\theta_{\text{pre}}}(t) + Q_{x,\text{Mask}}^{\theta}(t) \log \frac{Q_{x,\text{Mask}}^{\theta}(t)}{Q_{x,\text{Mask}}^{\theta_{\text{pre}}}(t)} \right\}$$

$$= \frac{\gamma}{T} \sum_{t=1}^{T} \sum_{\substack{x \in \mathcal{X} \\ x \neq \text{Mask}}} [\bar{x}_{t-1}^{(i)}]_x \left\{ -[\hat{x}_0^{\theta}]_x + [\hat{x}_0^{\theta_{\text{pre}}}]_x + [\hat{x}_0^{\theta}]_x \log \frac{[\hat{x}_0^{\theta}]_x}{[\hat{x}_0^{\theta_{\text{pre}}}]_x} \right\}$$

The simplified formula reduces the computational complexity of calculating $k^{(i)}(\theta)$ to $O(NT)$.

### E.3  SCHEDULE OF GUMBEL SOFTMAX TEMPERATURE

We use a linear schedule for the Gumbel softmax temperature $\tau$, decreasing over time as $\tau \sim 1/t$. In early time steps, the temperature is higher, introducing more uncertainty, while later steps have a lower temperature, approximating the true distribution more closely. This improves the fine-tuning procedure as the input becomes closer to clean data at later time steps and the uncertainty of model prediction is reduced.

## F  EXPERIMENTAL DETAILS AND ADDITIONAL RESULTS

In this section, we first provide a more detailed explanation of the baseline. We then discuss additional results and present more detailed settings for both regulatory DNA sequence design and protein sequence design. Finally, we present results of **DRAKES** in the natural language domain for toxicity-controlled text generation. We use 1 NVIDIA A100 80GB GPU for all experiments.

### F.1  BASELINES

In this section, we provide a detailed overview of each baseline method.

- **Guidance-based Methods.** Guidance-based methods are based on the pretrained model while adjusting during the sampling process according to the targeted property. This leads to longer inference time compared to fine-tuning approaches.
  - **CG** (Nisonoff et al., 2024). CG adjusts the transition rate of CTMC by calculating the predictor guidance:

    $$Q_{x,y|r}(t) = \frac{p(r|y,t)}{p(r|x,t)} Q_{x,y}(t)$$

    where $r$ is the target property, and the predictor guidance is further approximated using a Taylor expansion, i.e.

    $$\log \frac{p(r|y,t)}{p(r|x,t)} \approx (y-x)^T \nabla_x \log p(r|x,t)$$

    The predictor $p(r|x,t)$ is estimated using the posterior mean approach (Chung et al., 2022a), where the pretrained model is first utilized to estimate the clean data from the noisy input $x_t$, and then the reward oracle is applied to the predicted clean sequence. We remark that the above Taylor approximation doesn't have formal theoretical guarantees, considering that $x$ is discrete. This could be a reason why it does not work well in the case of protein-inverse folding in Section 6.3.

- **SMC** (Wu et al., 2024). SMC is a sequential Monte Carlo-based approach that uses the pretrained model as the proposal distribution. While it was originally designed for conditioning rather than reward maximization, it can be adapted for reward maximization by treating rewards as classifiers. In our experiment, we use this adapted version.
  - **TDS** (Wu et al., 2024). Similar to SMC, TDS also applies sequential Monte Carlo, but utilizes CG rather than the pretrained model as the proposal.

- **Classifier-free Guidance (CFG)** (Ho & Salimans, 2022). Unlike guidance-based methods, CFG trains a conditional generative model from scratch and does not rely on the pretrained model. To generate sequences $x$ with desired properties $r(x)$, CFG incorporates $r(x)$ as an additional input to the diffusion model and generates samples conditioning on high $r(x)$ values. Specifically, binary labels of $r(x)$ are constructed according to the $95\%$ quantile, and sampling is done conditioned on the label corresponding to high values of $r(x)$.

  It is important to note that CFG requires labeled data pairs $\{x, r(x)\}$ for training, which can limit its performance in cases with limited labeled data, especially when the pretrained model is already a conditional diffusion model $p(x|c)$. For example, in the protein inverse folding task, where $x$ is the protein sequence, $c$ is the protein structure, and $r(x)$ is the protein stability, CFG struggles, as shown in Table 2. This is due to the small size of the Megascale dataset (containing only a few hundred different protein structures), which reduces its capability and generalizability[2]. While data augmentation can be applied to construct additional training data, it is resource-intensive, requires significant case-by-case design, and is beyond the scope of this work. For the DNA sequence design task, since all sequences in the dataset are labeled, there is no such issue.

## F.2 REGULATORY DNA SEQUENCE DESIGN

In this section, we first outline the training process for reward oracles and pre-trained models in the regulatory DNA sequence design experiment. Subsequently, we provide additional explanations on fine-tuning setups and present further results.

**Reward Oracle.** We train reward oracles to predict activity levels of enhancers in the HepG2 cell line using the dataset from Gosai et al. (2023). Following standard practice (Lal et al., 2024), we split the dataset into two subsets based on chromosomes, with each containing enhancers from half of the 23 human chromosomes. We train two reward oracles on each subset independently using the Enformer (Avsec et al., 2021) architecture initialized with its pretrained weights. One oracle is used for fine-tuning, while the other is reserved for evaluation (i.e. *Pred-Activity* in Table 1). Denote the subset used for training the fine-tuning oracle as FT and the subset for training the evaluation oracle as Eval. Table 3 presents the model performance for both oracles on each subset. Both oracles perform similarly, achieving a high Pearson correlation ($> 0.85$) on their respective held-out sets (Eval for the fine-tuning oracle and FT for the evaluation oracle).

Table 3: Performance of the reward oracles for predicting HepG2 activity of enhancer sequences.

| Model | Eval Dataset | MSE ↓ | Pearson Corr ↑ |
|---|---|---|---|
| Fine-Tuning Oracle | FT | 0.149 | 0.938 |
| | Eval | 0.360 | 0.869 |
| Evaluation Oracle | FT | 0.332 | 0.852 |
| | Eval | 0.161 | 0.941 |

**Pretrained Model.** We pretrain the masked discrete diffusion model (Sahoo et al., 2024) on the full dataset of Gosai et al. (2023), using the same CNN architecture as in Stark et al. (2024) and a linear noise schedule. Other hyperparameters are kept identical to those in Sahoo et al. (2024). To assess the model's ability to generate realistic enhancer sequences, we sample 1280 sequences and compare them with 1280 randomly selected sequences from the original dataset. Figure 3 presents the distribution of HepG2 activity predicted by either the fine-tuning (FT) or evaluation (Eval) oracle for both the generated and original sequences, along with the true observations for the original sequences.

---

[2]In contrast, other methods (guidance-based methods and fine-tuning methods) leverage the pretrained model trained on the much larger PDB dataset ($\sim 23,000$ structures) and achieve better performance.

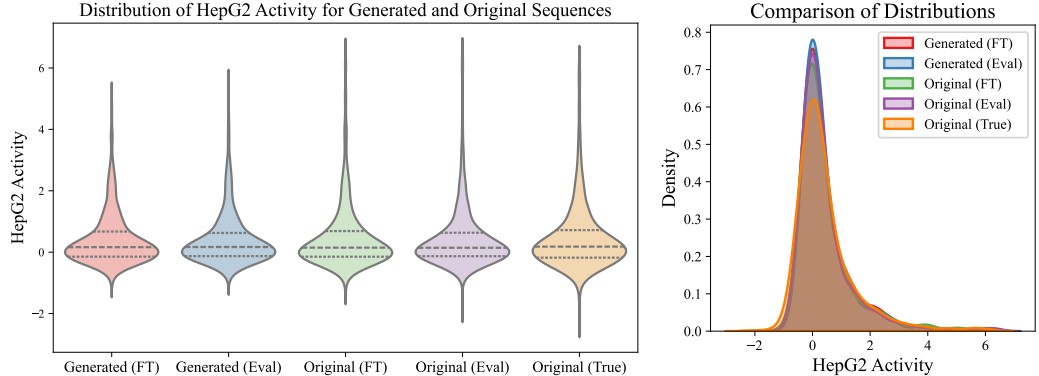

Figure 3: Comparison of HepG2 activity distributions between original sequences and those generated by the pretrained model. The activity distributions match closely with each other.

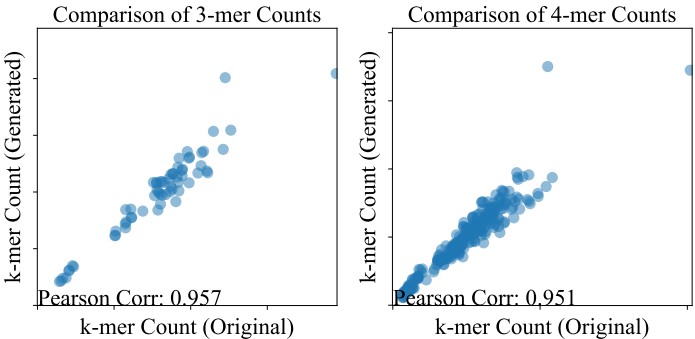

Figure 4: 3-mer and 4-mer Pearson correlation between the original and generated sequences.

The activity levels of the generated sequences align well with those of the original dataset, indicating the effectiveness of pretrained model in generating in-distribution enhancer sequences. Furthermore, Figure 4 shows the 3-mer and 4-mer Pearson correlation between the synthetic and original sequences, both of which exceed 0.95, further validating the model's performance.

**Fine-tuning Setup.** We utilize the pretrained masked discrete diffusion model and the fine-tuning oracle described above for fine-tuning. During **DRAKES**'s stage 1 data collection, sequences are generated from the model over 128 steps. We set $\alpha = 0.001$ to govern the strength of the KL regularization and truncate the backpropagation at step 50. The base Gumbel Softmax temperature is set to 1.0. The model is fine-tuned with 128 samples as a batch (32 samples per iteration, with gradient accumulated over 4 iterations) for 1000 steps. For **DRAKES** w/o KL, we follow the same setup, but set $\alpha$ to zero. For evaluation, we generate 640 sequences per method (with a batch size of 64 over 10 batches) for each random seed. We report the mean and standard deviation of model performance across 3 random seeds.

**Additional Results for Fine-Tuning.** Along with the median Pred-Activity values shown in Table 1, Figure 5 presents the full distribution of Pred-Activity for each method, which shows consistent patterns as Table 1.

**Ablation Study on Gradient Truncation Number.** To show the impact of the gradient truncation module, we conduct an ablation study on the gradient truncation number. As shown in Table 4, when the truncation number is small, the model cannot effectively update the early stage of the sampling process, leading to suboptimal performance. Meanwhile, when the truncation number is large, the memory cost of the model fine-tuning increases, and the optimization is harder due to the gradient accumulation through the long trajectory. Therefore, an intermediate level of gradient truncation leads to the best performance.

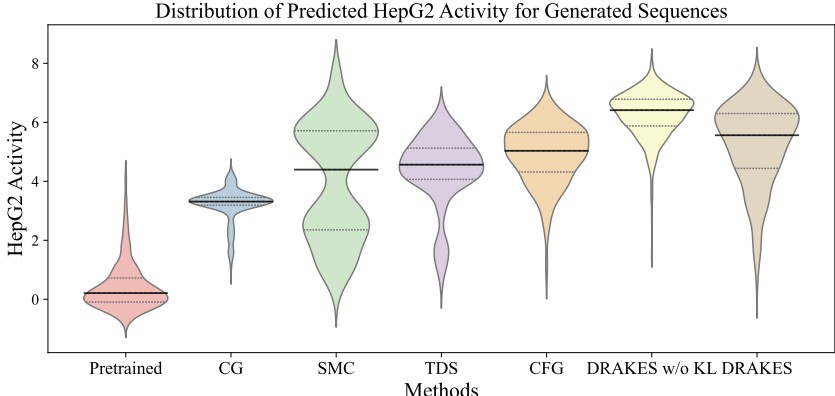

Figure 5: Distribution of Pred-Activity for the generated sequences of each method.

Table 4: Ablation study on gradient truncation number on regulatory DNA sequence design.

| Truncation Number | Pred-Activity (median)↑ | ATAC-Acc↑(%) | 3-mer Corr↑ | JASPAR Corr↑ | App-Log-Lik (median)↑ |
|---|---|---|---|---|---|
| 25 | 6.17 | 95.9 | 0.569 | 0.798 | -278 |
| **50** | 5.61 | 92.5 | 0.887 | 0.911 | -264 |
| 75 | 4.61 | 52.8 | 0.238 | 0.644 | -268 |

**Ablation Study on Gumbel Softmax Temperature Schedule.** We conduct an ablation study with different temperature schedules of Gumbel Softmax, i.e. a linear schedule and a constant schedule. As shown in Table 5, both schedules achieve similar performance, indicating the robustness of our method. In the performance reported in Table 1, we utilize the linear schedule.

Table 5: Ablation study on Gumbel Softmax temperature schedule on regulatory DNA sequence design.

| Temperature Schedule | Pred-Activity (median)↑ | ATAC-Acc↑(%) | 3-mer Corr↑ | JASPAR Corr↑ | App-Log-Lik (median)↑ |
|---|---|---|---|---|---|
| Constant | 5.89 | 91.6 | 0.852 | 0.914 | -258 |
| Linear | 5.61 | 92.5 | 0.887 | 0.911 | -264 |

## F.3 PROTEIN INVERSE FOLDING

We first discuss the setup of datasets used for training reward models. Next, we present the performance of the pre-trained model, followed by an evaluation of the reward models. Finally, we describe the fine-tuning setup and provide additional results.

**Dataset Curation.** We utilize the large-scale protein stability dataset, Megascale (Tsuboyama et al., 2023) for the protein inverse folding experiment, which contains stability measurements for $\sim 1.8$M sequence variants (for example, single mutants and double mutants) from 983 protein domains. We follow the dataset curation and train-validation-test splitting procedure from Widatalla et al. (2024). Specifically, the wild-type protein structures are clustered with Foldseek clustering and the data is split based on clusters. We then drop a few proteins with ambiguous wild type labels, and clip the $\Delta G$ values that are outside the dynamic range of the experiment ($> 5$ or $< 1$) to the closest measurable value (5 or 1) as in Nisonoff et al. (2024). We further exclude proteins where a significant proportion of the corresponding variants' $\Delta G$ measurements fall outside the experimental range. The final dataset consists of 438,540 sequence variants from 311 proteins in the training set, 15,182 sequences from 10 proteins in the validation set, and 23,466 sequences from 12 proteins in the test set.

**Pretrained Model.** We pretrain an inverse folding model using the discrete flow model loss from (Campbell et al., 2024) and the ProteinMPNN (Dauparas et al., 2022) architecture to encode both sequence and structure as model input. The model is trained on the PDB training set used in Dauparas et al. (2022), containing 23,349 protein structures and their ground truth sequences, which is distinct

from the dataset in Tsuboyama et al. (2023). We first evaluate the effectiveness of the inverse folding model on the PDB test set in Dauparas et al. (2022), which has 1,539 different proteins. As in Nisonoff et al. (2024), we set the temperature during sampling to be 0.1, and randomly sample one sequence conditioned on each structure for both our pretrained discrete flow model and the de facto inverse folding method, ProteinMPNN. As shown in Table 6, the pretrained model performs similarly to ProteinMPNN, achieving comparable sequence recovery rate.

Table 6: Model performance of protein inverse folding on PDB test set.

| Method | Sequence Recovery Rate (%) ↑ |
|---|---|
| ProteinMPNN | 47.9 |
| Discrete Flow Model | 48.6 |

We further evaluate the generalizability of the pretrained model to the proteins in the Megascale dataset. Results on both Megascale training and test set are shown in Table 7. We calculate the self-consistency RMSD (scRMSD) to assess how well a generated sequence folds into the desired structure. Specifically, the generated sequences are folded into 3D structures using ESMFold (Lin et al., 2023), and scRMSD is calculated as their RMSD relative to the original backbone structure we are conditioning on. An scRMSD lower than $2\mathring{A}$ is typically considered a successful inverse folding (Nisonoff et al., 2024; Campbell et al., 2024). As shown in Table 7, the pretrained model achieves a similar sequence recovery rate on Megascale as the PDB test set and low scRMSD, with a success rate greater than $90\%$, indicating its effectiveness on the inverse folding task.

Table 7: Model performance of protein inverse folding on Megascale proteins.

| Eval Dataset | Sequence Recovery Rate (%) ↑ | scRMSD ($\mathring{A}$) ↓ | %(scRMSD< 2)(%) ↑ |
|---|---|---|---|
| Megascale-Train | 47.0 | 0.825 | 95.0 |
| Megascale-Test | 44.0 | 0.849 | 90.9 |

**Reward Oracle.** We train the reward oracles on the Megascale dataset using the ProteinMPNN architecture. The oracles take both the protein sequence and the corresponding wild-type structure as input to predict the stability of the sequence, measured by $\Delta\Delta G$ (calculated as the difference in $\Delta G$ between the variant and the wild-type from the dataset). Similar to Nisonoff et al. (2024), the final layer of the ProteinMPNN architecture is mean-pooled and mapped to a single scalar with a 2-layer MLP, and the model weights before the mean-pooling are initialized with the weights from the pretrained inverse folding model.

Similar to the practice in the enhancer design experiment, we train two oracles – one for fine-tuning and one for evaluation. The fine-tuning oracle is trained on Megascale training set. We select the best epoch based on validation set performance, and report the Pearson correlation on both Megascale training and test set in Table 8. The performance gap between the training and test sets highlights the difficulty of generalizing to unseen protein structures in this task.

The evaluation oracle is trained on the complete dataset (train+val+test). To attain the best hyper-parameters, we randomly split the full dataset into two subsets, an in-distribution set for training, denoted as I, and an out-of-distribution set for validation, denoted as O. Note that here the evaluation oracle is trained part of the variants of *all* wild-type proteins (i.e. Megascale-Train-I & Megascale-Val-I & Megascale-Test-I), and the out-of-distribution set contains unseen sequence variants, but no new structures. The Pearson correlation on each subset is presented in Table 8. It achieves much higher correlations than the fine-tuning oracle, indicating good generalizability of the evaluation oracle to new sequences of in-distribution protein structures. For the final evaluation oracle used to calculate results in Table 2, we train it on the full dataset using the best hyperparameters selected as discussed. It achieves a Pearson correlation of 0.951 on Megascale training set and 0.959 on Megascale test set (both being in-distribution for the evaluation oracle).

**Finetuning Setup.** We utilize the pretrained inverse folding model and the fine-tuning oracle described above for fine-tuning. During **DRAKES**'s stage 1 data collection, we generate sequences from the model over 50 steps. We set $\alpha = 0.0003$ and truncate the backpropagation at step 25. The

Table 8: Performance of the reward oracles for predicting stability conditioned on protein sequence and structure, across a variety of Megascale subsets.

| Model | Eval Dataset | Pearson Corr ↑ |
|---|---|---|
| Fine-Tuning Oracle | Megascale-Train | 0.828 |
| | Megascale-Test | 0.685 |
| Evaluation Oracle | Megascale-Train-I | 0.948 |
| | Megascale-Train-O | 0.942 |
| | Megascale-Test-I | 0.955 |
| | Megascale-Test-O | 0.920 |

base Gumbel Softmax temperature is set to 0.5. The model is finetuned with proteins in Megascale training set with batch size 128 (16 samples per iteration, with gradient accumulated over 8 iterations) for 100 epochs. For **DRAKES** w/o KL, we follow the same setup, but set $\alpha$ to zero. The model is evaluated on Megascale test set, where we generate 128 sequences conditioned on each protein structure for every method (with a batch size of 16 over 8 batches) and each random seed. We report the mean and standard deviation of model performance across 3 random seeds.

**Evaluation Oracle Accounts for Over-Optimization.** As discussed in Section 6.2, for the enhancer design experiment, significant over-optimization occurs when evaluating Pred-Activity, even with an evaluation oracle trained on distinct data unseen during fine-tuning. In contrast, the protein inverse folding experiment largely mitigates this issue. Table 9 shows the median values of Pred-ddG for the generated sequences based on both the evaluation oracle (same as those reported in Table 2) and the fine-tuning oracle. Although **DRAKES** w/o KL shows significantly higher Pred-ddG than **DRAKES** with the fine-tuning oracle, their performance with the evaluation oracle remains similar, suggesting less pronounced over-optimization in evaluation. This is because enhancer sequences are relatively homogeneous, and even though we split based on chromosomes, each chromosome still has similar regions. However, protein structures are more distinct, and training on different proteins creates unique model landscapes.

Table 9: Model performance on protein inverse folding, with Pred-ddG calculated using either the evaluation oracle (Eval) or the fine-tuning oracle (FT).

| Method | Pred-ddG-Eval (median) ↑ | Pred-ddG-FT (median) ↑ |
|---|---|---|
| Pretrained | -0.544(0.037) | 0.161(0.012) |
| CG | -0.561(0.045) | 0.158(0.017) |
| SMC | 0.659(0.044) | 0.543(0.013) |
| TDS | 0.674(0.086) | 0.557(0.005) |
| CFG | -1.159(0.035) | -1.243(0.013) |
| **DRAKES** w/o KL | 1.108(0.004) | 0.833(0.000) |
| **DRAKES** | 1.095(0.026) | 0.702(0.002) |

**Ablation Study on Gradient Truncation Number.** To show the impact of the gradient truncation module, we conduct an ablation study on the gradient truncation number. The results are shown in Table 10. While the model performance is robust with different truncation numbers, an intermediate level of gradient truncation leads to the best performance.

Table 10: Ablation study on gradient truncation number on inverse protein folding.

| Truncation Number | Pred-ddG (median) ↑ | %(ddG> 0) (%) ↑ | scRMSD (median) ↓ | %(scRMSD< 2)(%) ↑ | Success Rate (%) ↑ |
|---|---|---|---|---|---|
| 15 | 0.977 | 83.2 | 0.852 | 92.4 | 76.0 |
| **25** | 1.095 | 86.4 | 0.918 | 91.8 | 78.6 |
| 35 | 1.033 | 84.8 | 0.868 | 92.7 | 77.7 |

**Ablation Study on Gumbel Softmax Temperature Schedule.** We conduct an ablation study with different temperature schedules of Gumbel Softmax, i.e. a linear schedule and a constant schedule.

As shown in Table 11, both schedules achieve similar performance, indicating the robustness of our method. In the performance reported in Table 2, we utilize the linear schedule.

Table 11: Ablation study on Gumbel Softmax temperature schedule on inverse protein folding.

| Temperature Schedule | Pred-ddG (median)↑ | %(ddG> 0) (%)↑ | scRMSD (median)↓ | %(scRMSD< 2)(%)↑ | Success Rate (%)↑ |
|---|---|---|---|---|---|
| Constant | 1.178 | 86.5 | 0.897 | 93.2 | 80.3 |
| Linear | 1.095 | 86.4 | 0.918 | 91.8 | 78.6 |

**Results with pLDDT.** In addition to scRMSD, we further measure the self-consistency of the generated sequences using pLDDT. The results are shown in Table 12. Following the common practice as in Widatalla et al. (2024), we utilize 80 as the cutoff threshold and define a success generation as having positive predicted stability (i.e., Pred-ddG>0) and confident ESMFold-predicted structure (i.e., pLDDT > 80). The results align well with those in Table 2 using scRMSD. **DRAKES** significantly outperforms all baseline methods in terms of the overall success rate.

Table 12: Model performance on inverse protein folding. **DRAKES** generates protein sequences that have both high stability and high confidence in ESMFold predicted structures, outperforming baselines in the overall success rate. We report the mean across 3 random seeds, with standard deviations in parentheses.

| Method | Pred-ddG (median)↑ | %(ddG> 0) (%)↑ | pLDDT (median)↑ | %(pLDDT> 80)(%)↑ | Success Rate (%)↑ |
|---|---|---|---|---|---|
| Pretrained | -0.544(0.037) | 36.6(1.0) | 87.9(0.0) | **87.5(0.1)** | 33.8(0.8) |
| CG | -0.561(0.045) | 36.9(1.1) | 87.9(0.0) | 87.5(0.2) | 34.3(0.5) |
| SMC | 0.659(0.044) | 68.5(3.1) | **88.3(0.1)** | 84.1(1.0) | 53.7(4.3) |
| TDS | 0.674(0.086) | 68.2(2.4) | 88.3(0.1) | 85.6(0.7) | 54.4(2.8) |
| CFG | -1.186(0.035) | 11.0(0.4) | 65.7(0.0) | 7.8(0.1) | 0.0(0.0) |
| **DRAKES** w/o KL | **1.108(0.004)** | **100.0(0.0)** | 73.4(0.2) | 40.4(0.2) | 40.4(0.2) |
| **DRAKES** | 1.095(0.026) | 86.4(0.2) | 87.0(0.0) | 85.6(0.7) | **72.3(0.8)** |

**Diversity of Generated Sequences.** We evaluate the diversity of the protein sequences generated by different methods using the average sequence entropy of each protein backbone in the test set. The results are shown in Table 13. **DRAKES** achieves a comparable level of diversity as the pretrained model, significantly outperforming SMC and TDS. This further justifies the efficacy of **DRAKES** to achieve the optimization objective of generating stable sequences, while maintaining high diversity. Among all baselines, **DRAKES** is the only method that have both a high success rate and high diversity.

Table 13: Average sequence entropy of the generated sequences on protein inverse folding.

| Method | Sequence Entropy ↑ |
|---|---|
| Pretrained | **34.7(0.2)** |
| CG | 34.6(0.1) |
| SMC | 24.9(1.2) |
| TDS | 24.9(0.5) |
| CFG | 8.4(0.1) |
| **DRAKES** w/o KL | 25.7(0.1) |
| **DRAKES** | 33.3(0.2) |

**Additional Results.** We provide more examples of the generated proteins in Figure 6, in addition to Figure 2. We also provide the specific values for energy, Pred-ddG and scRMSD of the visualized protein generated by **DRAKES**, as well as the energy values for the corresponding wild-type structure.

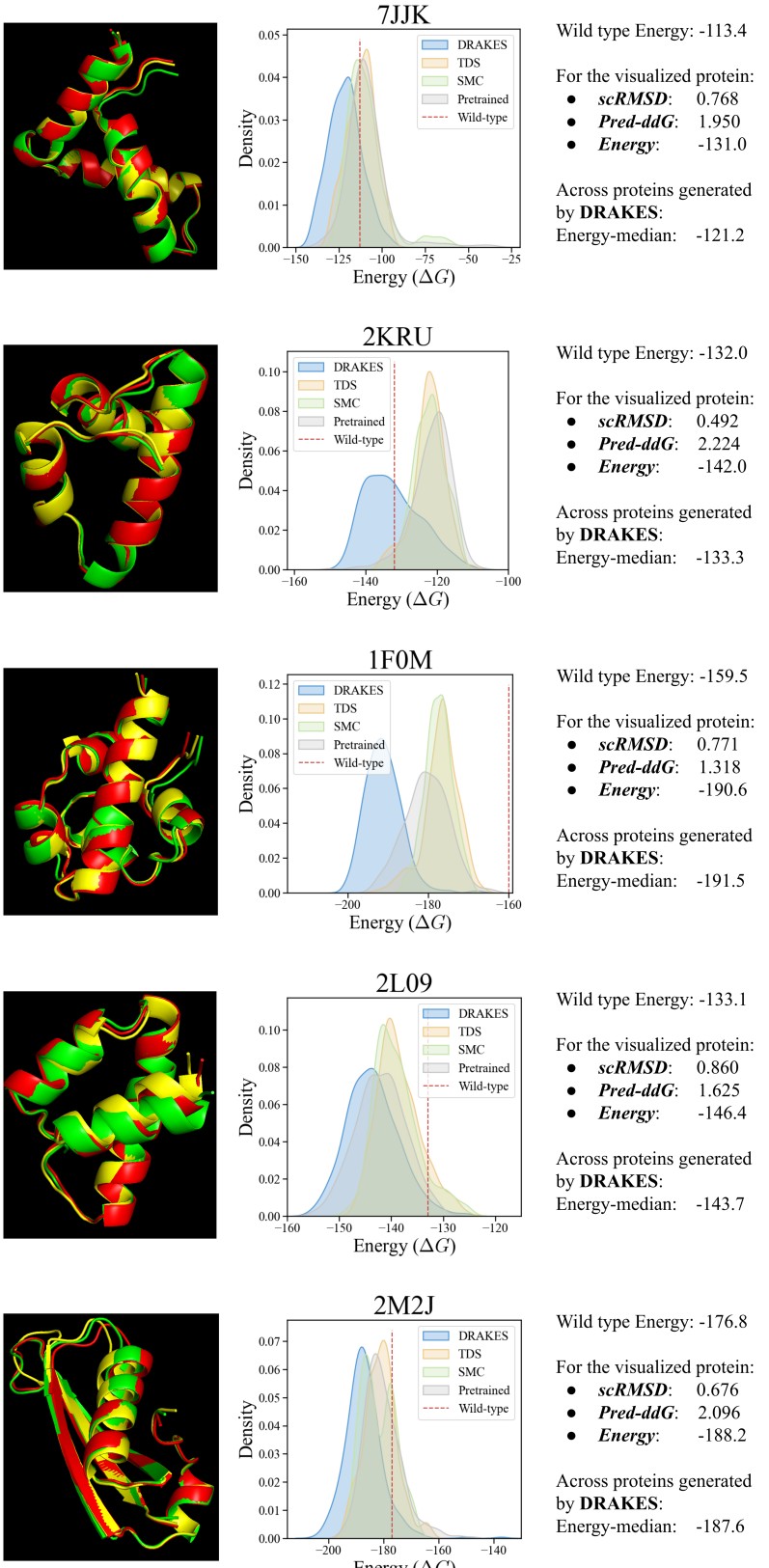

Figure 6: Additional examples of generated proteins.

### F.4 TOXICITY-CONTROLLED TEXT GENERATION

To demonstrate the general applicability of **DRAKES** beyond its use in the biological domain, we conduct experiments in toxicity-controlled text generation. The corresponding details and results are presented below.

**Reward Oracle.** We follow the setting in Zhao et al. (2024), using a toxicity prediction model trained by Corrêa (2023) as the reward oracle, which is a fine-tuned RoBERTa model with 125M parameters that can be used to score the toxicity of a sentence.

**Pretrained Model.** We utilize the pretrained masked diffusion language model from Sahoo et al. (2024) with 130M non-embedding parameters, which is trained on the OpenWebText corpus (Gokaslan et al., 2019) for one million steps and involves the processing of 33 billion tokens.

**Fine-tuning Setup.** Following (Zhao et al., 2024), we generate 20 output tokens. During **DRAKES**'s stage 1 data collection, sequences are generated from the model over 20 diffusion steps. We set $\alpha = 0.0001$ and truncate the backpropagation at step 5. The base Gumbel Softmax temperature is set to 1.0. The model is fine-tuned with batch size 128 (32 samples per iteration, with gradient accumulated over 4 iterations) for 100 steps. For **DRAKES** w/o KL, we follow the same setup, but set $\alpha$ to zero. For evaluation, we generation 320 sequences per method (with a batch size of 32 over 10 batches).

**Results.** As shown in Table 14, **DRAKES** generates sequences with the highest reward (i.e., low toxicity) compared to all baseline models, measured by the mean and median *Pred-Score*. This demonstrates the effectiveness of **DRAKES** in the natural language domain.

Table 14: Model performance on toxicity controlled text generation.

| Method | Pred-Score (mean)↑ | Pred-Score (median)↑ |
|---|---|---|
| Pretrained | 6.996 | 8.003 |
| CG | 8.488 | 9.009 |
| SMC | 9.286 | 9.367 |
| TDS | 9.817 | 9.882 |
| **DRAKES** | **10.044** | **10.049** |

