# OpenReview forum: "Fine-Tuning Discrete Diffusion Models via Reward Optimization with Applications to DNA and Protein Design"
_ICLR.cc/2025/Conference — ICLR 2025 Poster_

### Official Review · Reviewer_aA7C · 2024-10-24

**Soundness:** 3
**Presentation:** 3
**Contribution:** 2
**Rating:** 6
**Confidence:** 3

**Summary:**

The paper presents a novel method for fine-tuning discrete diffusion models using reward function by formulating it as a RL problem. To mitigate the challenge of non-differentiable trajectories, authors use gumbel-softmax trick that enables direct backpropagation of rewards. The proposed method conducts experiments on DNA and protein sequences and show effectiveness for generating both natural and high-rewarding sequences.

**Strengths:**

While most of diffusion fine-tuning methods conduct experiments on continuous spaces such as images, this paper tackles more practical and difficult setting, fine-tuning discrete diffusion models using reward function. The proposed method, gumbel-softmax trick for tackling non-differentiablility of discrete diffusion, is simple but have shown effectivenss in various experiments. Moreover, authors utilize various observations of prior methods for fine-tuning diffusion models, such as KL regularization from DPOK [1], truncated backpropagation from DRAFT [2].

[1] Fan, Ying, et al. "Reinforcement learning for fine-tuning text-to-image diffusion models." *Advances in Neural Information Processing Systems* 36.

[2] Clark, Kevin, et al. "Directly Fine-Tuning Diffusion Models on Differentiable Rewards." *The Twelfth International Conference on Learning Representations*.

**Weaknesses:**

**Gumbel softmax temperature)**

It seems that the temperature for the gumbel softmax, $\tau$ might be crucial for the performance. While the authors mentioned that linear schedule is deployed, it might be better to analyze the effect of temperature scheduling such as constant or loglinear.

**Position of the paper)**

The paper reminds me several similar papers… I noticed that fine-tuning discrete diffusion models is quite novel, but except for gumbel softmax trick (which is also widely used to differentiate discrete setting) other parts are really straightforward. It might be much better to propose a general framework for fine-tuning discrete diffusion models beyond biological sequences such as languages for more impactness. Otherwise, more comprehensive comparison with other models for biological sequence design can also make the paper to be unique. Nevertheless, I lean to acceptance.

**Questions:**

**Target Distribution)**

It is widely known that the target distribution of the following RL problem is proportional to $\exp(r(\cdot)/\alpha)p^{\text{pre}}(\cdot)$. However, recent works which formulates fine-tuning as stochastic optimal control state that naive KL regularization does not lead to the target distribution [1, 2]. Does this work not suffer from this issue?

**Comparison with other works)**

DNA Enhancers is a widely used benchmark for biological sequence designs. While the paper focus on discrete diffusion models, there are also works which fine-tune continuous diffusion models to generate biological sequences [3]. Is it better to fine-tune discrete diffusion models rather than continuous diffusion models for solving biological sequence problems? More analysis between those works may beneficial.

[1] Uehara, Masatoshi, et al. "Fine-tuning of continuous-time diffusion models as entropy-regularized control." *arXiv preprint arXiv:2402.15194* (2024).

[2] Domingo-Enrich, Carles, et al. "Adjoint matching: Fine-tuning flow and diffusion generative models with memoryless stochastic optimal control." *arXiv preprint arXiv:2409.08861* (2024).

[3] Uehara, Masatoshi, et al. "Bridging Model-Based Optimization and Generative Modeling via Conservative Fine-Tuning of Diffusion Models." *Advances in Neural Information Processing Systems* 37.

---

> ### Author Response · Authors · 2024-11-20
> **Response to Reviewer aA7C**
>
> We would like to thank the reviewer for their diligent and insightful review and their encouraging words. We have incorporated the reviewer’s comments in the updated paper, with the changes in the manuscript highlighted in magenta. We share our thoughts on the questions asked below.
>
>
> > Gumbel softmax temperature. It seems that the temperature for the gumbel softmax, $\tau$ might be crucial for the performance. While the authors mentioned that a linear schedule is deployed, it might be better to analyze the effect of temperature scheduling such as constant or loglinear.
>
>
> We conduct an ablation study with different temperature schedules of Gumbel Softmax, i.e. a linear schedule and a constant schedule. The results are shown in Table 5 and Table 11 in the updated manuscript, highlighted in magenta. Both schedules achieve similar performance, indicating the robustness and stability of our method.
>
>
> > Position of the paper. The paper reminds me several similar papers… I noticed that fine-tuning discrete diffusion models is quite novel, but except for gumbel softmax trick (which is also widely used to differentiate discrete setting) other parts are really straightforward. It might be much better to propose a general framework for fine-tuning discrete diffusion models beyond biological sequences such as languages for more impactness. Otherwise, more comprehensive comparison with other models for biological sequence design can also make the paper to be unique. Nevertheless, I lean to acceptance.
>
>
> The paper is motivated by and focuses on the practical applications of biological sequence design problems, a challenging and important application domain, as emphasized in the abstract and introduction sections. Our experimental settings on enhancer design and protein inverse folding are closely related to real-world gene therapies and protein-based therapeutics[a,b], and we provide rigorous and thorough evaluations on these settings.
>
> Additionally, we want to emphasize that biological sequences are the unique domains where masked diffusion models really shine due to the fact that they are really discrete (unlike image) and non-autoregressive (unlike text) by nature, which fits well with masked diffusion models. Indeed, the widely successful foundation model ESM3 in this field shares similarities to masked diffusion models in both training and inference (i.e., masked iterative decoding) procedures.
> Domains like text and pixel-level images are not the main focus of this paper.
>
> Further, our theory and method are novel in many aspects. We derive the first theoretical guarantee to address reward-maximizing finetuning in discrete diffusion models. This requires addressing unique challenges in the formulation of continuous-time Markov chains (CTMC), which differ from Brownian motion in continuous diffusion models. Also, the induced trajectories from CTMC are no longer differentiable, unlike in continuous spaces, which brings additional challenges. Additionally, our novel theoretical guarantee also establishes a connection with recent advancements in classifier guidance for discrete diffusion models.
>
> [a] Cell-type-directed design of synthetic enhancers. Nature 2024.
>
> [b] Aligning protein generative models with experimental fitness via direct preference optimization. bioRxiv 2024.

---

> ### Author Response · Authors · 2024-11-20
> **Response to Reviewer aA7C (continued)**
>
> > It is widely known that the target distribution of the following RL problem is proportional to $\exp(r(\cdot)/\alpha)p^{\text{pre}}(\cdot)$. However, recent works that formulate fine-tuning as stochastic optimal control state that naive KL regularization does not lead to the target distribution [1, 2]. Does this work not suffer from this issue?
> [1] Uehara, Masatoshi, et al. "Fine-tuning of continuous-time diffusion models as entropy-regularized control." arXiv preprint arXiv:2402.15194 (2024).
> [2] Domingo-Enrich, Carles, et al. "Adjoint matching: Fine-tuning flow and diffusion generative models with memoryless stochastic optimal control." arXiv preprint arXiv:2409.08861 (2024).
>
>
> As mentioned in Section 3.2, we use masked diffusion models for this paper and the initial distribution is a Dirac delta distribution (completely masked state) following existing works of masked diffusion models. In this case, there is no initial bias problem. We have updated the paper to further clarify this. Note that masked diffusion models are broadly utilized for discrete diffusion models ([c,d]), and show much stronger performance than uniform prior as the results in SEDD[e] in text generation.
>
> If we extend our setting where the initial distribution is not a Diract delta, the initial value bias problem arises. We can expand it by learning an initial distribution again as in [f] or changing a masking schedule as in [g]. We have added this extension to our settings in the updated paper.
>
> [c] Simple and effective masked diffusion language models. NeurIPS 2024.
>
> [d] Simplified and generalized masked diffusion for discrete data. NeurIPS 2024.
>
> [e] Discrete diffusion language modeling by estimating the ratios of the data distribution. ICML 2024.
>
> [f] Fine-tuning of continuous-time diffusion models as entropy-regularized control. arXiv 2024.
>
> [g] Adjoint matching: Fine-tuning flow and diffusion generative models with memoryless stochastic optimal control. arXiv 2024.
>
>
>
> > Comparison with other works. DNA Enhancers is a widely used benchmark for biological sequence designs. While the paper focuses on discrete diffusion models, there are also works that fine-tune continuous diffusion models to generate biological sequences [3]. Is it better to fine-tune discrete diffusion models rather than continuous diffusion models for solving biological sequence problems? More analysis between those works may be beneficial.
> [3] Uehara, Masatoshi, et al. "Bridging Model-Based Optimization and Generative Modeling via Conservative Fine-Tuning of Diffusion Models." Advances in Neural Information Processing Systems 37.
>
>
> Thank the reviewer for the suggestions. Our focus of this paper is on discrete diffusion models. While it is interesting to compare diffusion models in the continuous space like Dirichlet diffusion used in [3] versus discrete diffusion, this is out of the scope of this paper and a potential future direction.

---

> > ### Comment · Reviewer_aA7C · 2024-11-26
> >
> > Thanks to the authors for providing additional experiments and clarifications, especially on the initial bias problem.
> > While I'm still concerned that the comparison with fine-tuning continuous diffusion models for biological sequence design is required and is not out of scope as both works try to solve the same problem, I keep maintaining a positive score.

---

> ### Author Response · Authors · 2024-11-28
> **Response to the Latest Comment by Reviewer aA7C**
>
> We thank the reviewer for carefully reading and helping us improve our manuscript, and for the positive view on our paper. Regarding comparison with fine-tuning continuous diffusion models for biological sequence design, we agree that it is an important and interesting question, while concerning that a fair comparison is hard to obtain since the base pretrained models themselves could differ in performance significantly. That's why we fix pretrained discrete diffusion models to make a fair comparison over fine-tuning methods.

---

### Official Review · Reviewer_X1rg · 2024-11-01

**Soundness:** 3
**Presentation:** 2
**Contribution:** 3
**Rating:** 6
**Confidence:** 3

**Summary:**

The authors propose a new methodology for fine-tuning discrete diffusion models in a simulation-based manner by back-propagating through the forward generative dynamics. They sidestep the problem of discrete sampling by leveraging the Gumbel based soft approximator for categorical sampling by first replacing the sampling with Gumbel-max and then performing the approximation by replacing this max operator with a softmax. The proposed algorithm, DRAKES, shows improved benefits in the domain of DNA sequences, where fine-tuning is with respect to a reward to optimize for activity of the sequences. The authors also conduct experiments and show improved performance on protein sequence design, especially inverse folding models to optimize for stable sequence generation. Overall, the authors show that their KL-regularized objective yields significant improvements when finetuned on a downstream reward, and without this KL regularization they often end up with over-optimized solutions, often lacking diversity and being unrealistic, especially under the pre-trained discrete diffusion model.

**Strengths:**

- The proposed work is well motivated theoretically and tackles a challenging problem of fine-tuning discrete diffusion models to allow for realistic generation while respecting certain desirable downstream properties, defined by a reward model.
- The algorithm, DRAKES, outperforms some of the baselines considered in the work (eg. guidance and SMC-based approaches) and shows significant promise in terms of generating realistic protein and DNA sequences while optimizing for better stability / activity.
- While similar approaches have been considered in continuous space diffusion models, this work tackles the challenging problem of discrete space. In particular, it constructs theory to be able to formally do this finetuning and shows alternate ways of looking at guidance approaches already proposed for discrete space models.

**Weaknesses:**

- I found the draft a bit hard to follow. If I understood it correctly, the preliminary section deals with generation from mask to noise as a trajectory from $t = T\ldots 0$, whereas this story is flipped from Section 3.2 onwards where the generation is now considered as a process from $t = 0\ldots T$.
- In line with the above problem, it was quite confusing to switch between $Q_{x, y}$ to $Q_{y, x}$, without really having some notion of what the $(i, j)$ entry of $Q$ represents. This might be because of my lack of understanding, but does it mean that $p(x_{t+dt} = i | x_t = j) = \sum_{k} Q_{i, k} p(x_t = k)$?
- The authors use truncated backpropagation to train their setup and the bias introduced due to that is unclear. Can the authors provide some kind of ablation into how the amount of truncation affects task performance?
- For the equation under Section 5.2, the left hand side of the equation is a function of $x$ and $t$ whereas the right side of the equation is not. Is there a typo?
- It is not quite clear whether the theory proposed in the work is very novel and insightful or a direct extension of existing theory in Uehara et. al 2024, for example. In particular, could the authors clarify that in their setup, what exactly is different and how is it not covered in prior work?

**Questions:**

- Could the authors write Equation 3 in terms of the KL divergence terms, to better understand what is the exact KL divergence that they want to regularize with. It is not immediately clear where the terms not associated with the $\log$ come from.
- For the equation under Stage 1, how does the probability that the authors write sum to 1? In particular, the sum over y of that equation would be greater than or equal to 1 always, unless $Q$ is negative, which will lead to other issues.
- Could the authors make sure that they have an equation number for all of their equations?

---

> ### Author Response · Authors · 2024-11-20
> **Response to Reviewer X1rg**
>
> We are grateful to the reviewer for the time they put in to review our work. We are glad to see that they recognize several strengths in our work, including the theoretical contributions of our paper and the superior empirical performance of our method. Below, we share our thoughts on the questions asked, with **(a) a thorough explanation of the novelty of our theory, (b) an ablation study on truncated backpropagation, and (c) improved notations**. We have added further clarifications in the updated manuscript, with changes highlighted in magenta.
>
>
> > I found the draft a bit hard to follow. If I understood it correctly, the preliminary section deals with generation from mask to noise as a trajectory from t=T…0, whereas this story is flipped from Section 3.2 onwards where the generation is now considered as a process from t=0…T.
>
>
> Yes, your observation is correct. From section 3.2., to simplify the notation, we go from $t=0$ to $t=T$ (otherwise, we always need to use $T-t$ instead of $t$). We have added this clarification in the updated manuscript.
>
>
> > In line with the above problem, it was quite confusing to switch between $Q_{x,y}$ to $Q_{y,x}$, without really having some notion of what the $(i, j)$ entry of $Q$ represents. This might be because of my lack of understanding, but does it mean that $p(x_{t+dt}=i|x_t=j)=\sum_k Q_{i,k} p(x_t=k)$?
>
> Thank the reviewer for pointing this out. Yes, there was some confusion in the original paper with the switching between $Q_{x,y}$ and $Q_{y,x}$. We have unified the notations and added clarifications to the meaning of $Q_{x,y}$ (i.e., ***$Q_{x,y}$ represents the transition rate matrix from state $x$ to state $y$***) in the updated paper, highlighted in magenta.
>
> We follow the convention of continuous-time Markov Chain (CTMC) parametrization in [a,b]. To be specific, $Q_{x,y}$ is the infinitesimal transition rate matrix from state $x$ to state $y$, with $Q_{x,x}=-\sum_{y \neq x} Q_{x,y}$ holds. Note that the rate matrix $Q$ is different from the transition probability $p$. The transition probability $p(x_{t+dt}=i|x_t=j)$ can be expressed using the rate matrix $Q$ as $p(x_{t+dt}=i|x_t=j)=Q_{j,i} \cdot dt$ for $i \neq j$, and $p(x_{t+dt}=i|x_t=j)=1 - \sum_{k \neq j} Q_{j,k} \cdot dt$ for $i = j$.
>
>
> [a] Convergence Analysis of Discrete Diffusion Model: Exact Implementation through Uniformization. arXiv 2024.
>
> [b] Discrete Diffusion Modeling by Estimating the Ratios of the Data Distribution. ICML 2024.
>
>
> > The authors use truncated backpropagation to train their setup and the bias introduced due to that is unclear. Can the authors provide some kind of ablation into how the amount of truncation affects task performance?
>
> Thank the reviewer for the suggestions. We have included an ablation study on the gradient truncation number to show the impact of the gradient truncation module in the updated manuscript (Table 4 for the enhancer design task and Table 10 for the protein inverse folding task), as highlighted in magenta.
>
> When the truncation number is small, the model cannot effectively update the early stage of the sampling process, leading to suboptimal performance. Meanwhile, when the truncation number is large, the memory cost of the model fine-tuning increases, and the optimization is harder due to the gradient accumulation through the long trajectory. Therefore, an intermediate level of gradient truncation leads to the best performance. Similar gradient truncation tricks have been applied in fine-tuning algorithms for continuous diffusion models and proved effective in practice [c,d].
>
> [c] Directly Fine-Tuning Diffusion Models on Differentiable Rewards. ICLR 2024.
>
> [d] Video Diffusion Alignment via Reward Gradients. arXiv 2024.

---

> ### Author Response · Authors · 2024-11-20
> **Response to Reviewer X1rg (continued)**
>
> > For the equation under Section 5.2, the left hand side of the equation is a function of x and t
>  whereas the right side of the equation is not. Is there a typo?
>
> The right-hand side of the equation means optimizing with respect to the function group $g:\mathcal{X} \rightarrow \mathbb{R}$, where the optimal candidate corresponds to the exponential value function on the left-hand side. Thus, this is an equation at the function level. We have updated the equation in the revised manuscript, as highlighted in magenta.
>
> > It is not quite clear whether the theory proposed in the work is very novel and insightful or a direct extension of existing theory in Uehara et. al 2024, for example. In particular, could the authors clarify that in their setup, what exactly is different and how is it not covered in prior work?
>
> We clarify the novelty compared to Uehara et.al 2024.
>
> 1) Results are for diffusion models in continuous states: Uehara et al. 2024. Their formulation is within continuous-time Brownian motion. We consider diffusion models for discrete states. We use a continuous time Markov chain (CTMC), which is completely different from Brownian motion. At this point, we would say the setups are very different as Brownian motion is heuristically seen as a sequence of Gaussian distribution while CTMC is seen as a sequence of categorical distribution.
> 2) In the KL term in (3), we calculate the path measure between the distribution induced by pre-trained models and fine-tuned models in CTMC. This way and form are completely different from the one in Brownian motion.
> 3) The proof and intermediate lemmas to prove the main statement are significantly different. For example, as an important immediate result, we derive the optimal generator in Theorem 2 in Section 5.1. As a corresponding result, in Lemma 2, Uehara et al. derive the optimal drift term in Brownian motion. These two take entirely different forms.
> 4) We have an insightful result unique to discrete diffusion models. An intermediate result in our proof, which is highlighted in bullet point 3, can also derive a classifier guidance algorithm tailored to diffusion models. This takes a completely different form compared to classifier guidance in diffusion models. We have highlighted this interesting observation in Section 5.1.
>
> > Could the authors write Equation 3 in terms of the KL divergence terms, to better understand what is the exact KL divergence that they want to regularize with. It is not immediately clear where the terms not associated with the log come from.
>
> The KL term for CTMC has exactly the formula of the whole second term in Equation 3 (Equation 5 in the updated paper) as highlighted in curly brackets in the equation, which is formally proved in Proposition 2 of [a].
>
> [a] Convergence Analysis of Discrete Diffusion Model: Exact Implementation through Uniformization. arXiv 2024.
>
> > For the equation under Stage 1, how does the probability that the authors write sum to 1? In particular, the sum over y of that equation would be greater than or equal to 1 always, unless
> Q is negative, which will lead to other issues.
>
> In continuous-time Markov Chain (CTMC) parametrization, $Q_{x,y}$ is the infinitesimal transition rate matrix from state $x$ to state $y$, with $Q_{x,x}=-\sum_{y \neq x} Q_{x,y}$ holds. Note that the rate matrix $Q$ is different from the transition probability $p$. Therefore, the probability in the equation is guaranteed to sum to 1. This is also noted in other papers, such as Equation 2 in Section 2.1 of [b].
>
> [b] Discrete Diffusion Modeling by Estimating the Ratios of the Data Distribution. ICML 2024.
>
> > Could the authors make sure that they have an equation number for all of their equations?
>
> Thank the reviewer for the suggestion. We have included the equation number for all equations in the updated manuscript.

---

> ### Comment · Reviewer_X1rg · 2024-11-25
> **Reviewer Response**
>
> Thanks to the authors for providing additional clarifications regarding their approach. I had a couple additional concerns that I wanted to follow up on, which are explained below.
>
> **Relevance of Theory**. While the authors' response regarding the existing theory for continuous space models in Uehara et.al 2024 was helpful, it is still not clear to me what the novelty of Theorem 1 is. In particular, for any problem of the form
> \begin{align}
> \arg\max_q \mathbb{E}_{q}[r(x_T)] - \alpha \mathbb{KL}[q || p]
> \end{align}
> this optimization procedure can be written as
> \begin{align}
> \arg\min_q \mathbb{E}_q [\log \frac{q(x\_{0:T})}{p(x\_{0:T})e^{r(x\_{T})/\alpha}}]
> \end{align}
> Suppose $Z = \int p(x\_{0:T})e^{r(x\_{T})/\alpha} dx\_{0:T}$. Then the above optimization can be written as
> \begin{align}
> \arg\min_q \mathbb{KL} [q(x\_{0:T}) || \frac{p(x\_{0:T})e^{r(x\_{T})/\alpha}}{Z}] - \log Z
> \end{align}
> This optima is clearly obtained when $q = \frac{p(x\_{0:T})e^{r(x\_{T})/\alpha}}{Z}$, or in other words, $q \propto p(x\_{0:T})e^{r(x\_{T})/\alpha}$. Maybe there is something wrong in the argument I am making here, and it would be very helpful if the authors could clarify what is different in their setup than the above, which should hold for both continuous and discrete random variables.
>
> **RTB Baseline**. I appreciate the authors for running this very relevant baseline. In case I am missing something, the baseline is trained with the following objective
> \begin{align}
> \left(\log Z_\phi + \sum \log p_\phi^{post}(x_t | x_{t-1}) - \sum \log p_\theta^{prior}(x_t | x_{t-1}) - \log R(x_T) \right)^2
> \end{align}
> Going by the authors' rebuttal code; in particular line $46$ of drakes_dna/rebuttal_finetune_rtb.py, it seems that they have interchanged the position of the posterior probability and the prior probability of the trajectory. Additionally, correct me if I am wrong but line 47 tries to approximate the term $Z_\phi$ with an importance sampling / empirical mean estimate but the sign for it looks wrong.
>
> In conclusion, I thank the authors for providing additional results ablating over back-propagation truncation as well as clarity in both writing as well as about the rate matrix. I would really appreciate further clarity on the points raised above; for which I could very well be mistaken.

---

> ### Author Response · Authors · 2024-11-27
> **Response to the Latest Comment by Reviewer X1rg**
>
> We thank the reviewer for their time and feedback. We provide our response to each of the additional concerns below.
>
> * **Relevance of Theory**
>
> Thank the reviewer for bringing this important up. We agree with your derivation in a discrete-time formulation. While we acknowledge it, as we cited Ziegler et al. 19.[1], we consider the continuous-time formulation, and we need to consider additional more non-trivial aspects, like KL divergence for path measures in CTMC, and how to go from discrete-time to continuous-time formally under realizability (well-specification) for generator matrices (but not like under realizability of the whole trajectories in the derivation you wrote in a discrete-time formulation). That’s why we invoke theorems such as Kolmogorov forward and backward equations in CTMC.
>
> That being said, the reviewer’s claim could be rephrased as a broader question in diffusion models: why do we need conditions-time formulation in diffusion models beyond discrete-time formulation? As a general answer, the continuous-time formulation gives a unified framework, flexibility in time steps, usage of efficient sampling schemes, and a better theoretical understanding of the connection to ratio matching, etc. [2,3]. In addition,  in our context,  it opens up the derivation of inference-time methods tailored to discrete diffusion models, which is hard to derive from a discrete-time perspective, as discussed in Section 5.2.
>
> [1] Fine-tuning language models from human preferences. arXiv 2019.
>
> [2] Discrete diffusion language modeling by estimating the ratios of the data distribution. ICML 2024
>
> [3] A continuous time framework for discrete denoising models. NeurIPS 2022.
>
> * **RTB Baseline**
>
> Thank the reviewer for pointing this out. We run experiments with the updated code. (Note that regarding the second point in line 47, we might be missing something but it appears to be a bug in RTB’s initial source code: https://github.com/GFNOrg/diffusion-finetuning/blob/main/diffusion_lm/finetune_cond_data.py#L154.)
>
> However, we still cannot achieve good performance (the maximal average reward it can achieve is around 0.7, which is slightly higher than the pretrained model (~0.3), while significantly lower than all other baselines).

---

> > ### Comment · Reviewer_X1rg · 2024-11-28
> > **Reviewer Response**
> >
> > Thanks for providing clarifications regarding the theory. Additionally, regarding the RTB baseline, did the authors run the baseline with the sign flipped for line 47 or without (if without, they should consider running it with the sign flipped)? Additionally, to which table in the paper should I be comparing these numbers against?

---

> ### Author Response · Authors · 2024-11-28
> **Response to the Latest Comment by Reviewer X1rg**
>
> Thank the reviewer for the question. The RTB baseline is run with the sign in line 47 flipped (the code has been updated in the anonymous link). The numbers correspond to the results in Table 1, column "Pred-Activity". (Note that the results in Table 1 are the median across generated sequences while the numbers mentioned in the rebuttal are the mean. Nevertheless, they have similar values.) All methods in Table 1 except for the pretrained model have Pred-Activity larger than 3.

---

> > ### Comment · Reviewer_X1rg · 2024-11-29
> > **Reviewer Response**
> >
> > Thanks for providing further clarifications. The authors have clarified all my concerns, and therefore I have increased my score.

---

> ### Author Response · Authors · 2024-12-02
> **Thanks to Reviewer X1rg**
>
> We thank the reviewer for investing their time and their reassessment and acknowledgment of our detailed explanations, along with increasing their score. We are pleased to hear that these clarifications have addressed all of their concerns, and we will revise the manuscript based on our discussions. We truly appreciate the reviewer’s effort in helping us refine this draft.

---

### Official Review · Reviewer_1d7y · 2024-11-03

**Soundness:** 2
**Presentation:** 2
**Contribution:** 2
**Rating:** 6
**Confidence:** 5

**Summary:**

This paper introduces DRAKES an approach to performing fine-tuning for discrete diffusion models. In particular, the approach is predicated on maximizing the expected reward of a trajectory that is controlled with a KL constraint to prevent gaming the reward function itself. The authors suggest using the Straight-Through Gumbel Softmax estimator to backpropagate the non-differentiable loss function. Experiments are conducted on biological sequence design tasks such as inverse folding with stability. Empirical results demonstrate that DRAKES can indeed improve reward metrics in the considered settings.

**Strengths:**

The approach considered in this paper is quite logical and follows a general approach of MaxEnt RL applied to fine-tuning generative models. In some sense, there is a tight connection to existing RLHF approaches to fine-tuning in the autoregressive setting. As a result, it is encouraging to see that the rewards do end up improving with the considered objective for discrete diffusion models in the smaller-scale biological sequence design tasks considered in this paper.

**Weaknesses:**

Despite some strengths in this paper, I share several concerns regarding the method, its scalability, and evaluation protocol. I will outline this below:

**Theoretical questions**

So I believe the result in Theorem 3 is not true and needs an additional assumption. I will give an intuitive argument as to why this might be the case and I am open to being wrong here. If I am indeed wrong I would love for the authors to correct and fix my understanding. Essentially, I believe there is an initial value bias problem as outlined in "Adjoint Matching" (Domingo-Enrich 2024) and the initial starting state affects the optimal fine-tuning distribution you hit when you add the KL regularization term. While they formulated this result in the continuous setting, I believe it equally holds in the discrete setting because it relies on simple facts of the continuity (Kolmogorov forward/backward) equations. More particularly, this fact arises due to the usage of not using a memory-less noise schedule. In the discrete setting, the masked prior is likely fine but say using a uniform or another prior would make this problematic. Currently, all the theory in this paper attempts to be more general and this is where things I suspect break, while for the masked prior we automatically achieve the memory-less property.

**Technical concerns**

DRAKES as a method requires rollouts of entire trajectories and bears resembles to the method Relative Trajectory Balance (Venkataramman et. al 2024) eq 12. In fact, it is largely the same except that in DRAKES Gumbel-softmax is used instead of learning the partition function. This also means that DRAKES is not scalable because there is an expensive process of collecting full trajectories and doing stochastic backdrop through the trajectory. This induces a bias that is accumulated through the length of the trajectory. This would suggest that the current method has difficulty scaling to longer sequences and larger problem domains. This is perhaps evidenced by the fact the authors do not consider fine-tuning on more standard discrete domains like text, or pixel-level fine-tuning.

Regarding, the Gumbel-Softmax estimator I would encourage the authors to try more modern gradient estimators. One suggestion is REINMAX (Liu et. al 2023). I believe this alleviates some of the bias and is more scalable than Gumbel-softmax. I also encourage the authors to do a bit more theoretical analysis of the error incurred by using
 as all the theory relies on not doing the Gumbel-softmax approximation so doesn't apply to the practical settings considered in the paper.

**Experimental questions**

I have some further questions and concerns regarding the considered experiments. In particular, the protein stability experiments do not indicate what length protein sequences are used. I suspect, the preprocessing of PDB largely restricts the protein sequences to small ones which would be in line with my estimation that the method does not scale to larger sequences. Moreover, it is surprising why pLDDT is not reported as a metric. This is the most common metric to show that protein sequences are designable. Self-consistency makes more sense in settings where structure is generated.

Moreover, I would like to see a few more metrics that characterize the diversity of generated samples pre and post fine-tuning with DRAKES for protein sequences. Something like the number of clusters as I suspect there is likely a sharp decline in diversity which of course would not be captured by self-consistency.

At present, DRAKES is presented as a general-purpose fine-tuning approach to discrete diffusion models but is only tested in biological sequence design. Nothing about DRAKES suggests it would not be feasible to attempt more standard discrete generative modeling settings. As such, I would like to see some standard text benchmarks for example fine-tuning for sentiment as done in Twisted SMC (Zhao et. al 2024) or class conditional fine-tuning in pixel-level image modeling. These settings would help understand both quantitatively and qualitatively, through generated samples, the empirical caliber of DRAKES.

Given the suspected similarity of DRAKES and RTB I would encourage the authors to include RTB as a baseline in there experimental settings.

**Presentation weakness**

All of the theory in the paper is rather incomplete and just plain sloppy. The appendices do not have complete proofs or the exact theorem statements. There is an air of informality that is scary because I believe some of the statements are actually not totally accurate and need a few assumptions as outlined previously.

Lemma 1 & 2 proofs are not complete. You cannot just skip the converse part. This is not a complete proof and is a bit sloppy. I encourage the authors to complete the proof.

Log Likelihood in Table 1 is actually misleading. You are likely computing an upper bound to Perplexity as done in MDLM.

**Typos and minor details**

Line 848 "competed" -> completed
Concrete distribution (Maddison et. al 2016) should be cited for Gumbel-softmax as well.


**Concluding remarks**

I am open to increasing my score if all the points in my review are sufficiently addressed, especially the experimental ones.

**References**

Liu, Liyuan, et al. "Bridging discrete and backpropagation: Straight-through and beyond." Advances in Neural Information Processing Systems 36 (2024).

Venkatraman, Siddarth, et al. "Amortizing intractable inference in diffusion models for vision, language, and control." arXiv preprint arXiv:2405.20971 (2024).

Zhao, Stephen, et al. "Probabilistic inference in language models via twisted sequential monte carlo." arXiv preprint arXiv:2404.17546 (2024).

**Questions:**

I encourage the authors to respond to my theory questions on whether there is indeed an initial value bias problem for prior distributions outside of the masked prior.

---

> ### Author Response · Authors · 2024-11-20
> **Response to Reviewer 1d7y**
>
> We thank the reviewer for their time and feedback that helped us improve the work.  We believe that there are a few confusions regarding our method, motivations, and experimental results that have resulted in the given rating. We have endeavored to address your concerns as concretely as possible and ask for your careful consideration of our clarifications. All of the discussions below are added to the revised manuscript in magenta.
>
>
> **Theoretical questions**
>
> > The result in Theorem 3 is not true and needs an additional assumption.
>
> We are confident that our statement is correct. As mentioned in Section 3.2, we use masked diffusion models for this paper and explicitly mention that the initial distribution is a Dirac delta distribution (completely masked state) following existing works of masked diffusion models. In this case, there is no initial bias problem. We have updated the paper to clarify this further. Note that masked diffusion models are broadly utilized for discrete diffusion models ([a,b]), and show much stronger performance than uniform prior as the results in SEDD[c] in text generation.
>
> If we extend our setting where the initial distribution is not a Diract delta, the initial value bias problem arises. We can expand it by learning an initial distribution again as in [d] or changing a masking schedule as in [e]. We will add this extension to our setting in the updated paper.
>
> [a] Simple and effective masked diffusion language models. NeurIPS 2024.
>
> [b] Simplified and generalized masked diffusion for discrete data. NeurIPS 2024.
>
> [c] Discrete diffusion language modeling by estimating the ratios of the data distribution. ICML 2024.
>
> [d] Fine-tuning of continuous-time diffusion models as entropy-regularized control. arXiv 2024.
>
> [e] Adjoint matching: Fine-tuning flow and diffusion generative models with memoryless stochastic optimal control. arXiv 2024.

---

> > ### Comment · Reviewer_1d7y · 2024-11-23
> > **Re: Theory Response**
> >
> > I thank the authors for their rebuttal and their attempts to respond to my theoretical questions. I believe your theory in the specific case of masked diffusion is true. I was not questioning this aspect. I was questioning the application of this to discrete diffusion models beyond a masked prior. In section 3.2 you indeed consider a masked prior, but the theoretical statements do not make this explicit. My direct request was to polish the theoretical statements and actually state the main assumption that you start from the masked prior.
> >
> > Also, your understanding is probably correct but phrased awkwardly. Masked diffusion models do not have the initial value bias problem not because they start from a Dirac state, but rather because they satisfy the memory-less property. Intuitively, the starting prior has no information on the target hit because all chains start from an absorbing state. Contrast this to a uniform prior where u could place a Dirac mass on points sampled from a uniform (Dirichlet) distribution on the Simplex and learn paths to a target. In this case, only the memory-less noise schedule achieves the desired property.

---

> ### Author Response · Authors · 2024-11-20
> **Response to Reviewer 1d7y (continued)**
>
> **Technical concerns**
>
> > DRAKES as a method requires rollouts of entire trajectories and bears resembles to the method Relative Trajectory Balance (Venkataramman et. al 2024) eq 12. In fact, it is largely the same except that in DRAKES Gumbel-softmax is used instead of learning the partition function. This also means that DRAKES is not scalable because there is an expensive process of collecting full trajectories and doing a stochastic backdrop through the trajectory. This induces a bias that is accumulated through the length of the trajectory. This would suggest that the current method has difficulty scaling to longer sequences and larger problem domains. This is perhaps evidenced by the fact the authors do not consider fine-tuning on more standard discrete domains like text, or pixel-level fine-tuning.
>
>
> First, we want to emphasize the key differences between our method and RTB, which is also included in the updated paper highlighted in magenta.
>
> 1) It appears that RTB[f] uses a trajectory balance loss.  On the other hand, we solve a control problem with direct backpropagation. Hence, the actual algorithms are significantly different.
> 2) Theoretically, our result has more additional insight tailored to discrete diffusion models. As an intermediate result to show our main claim Theorem 1, we show that the optimal generator in our control problems can derive non-trivial classifier guidance in discrete diffusion models (Theorem 2 in Section 5.1.). While [f] has a certain discussion around the connection with classifier guidance in the diffusion models for continuous states, they don’t show the above insightful result for diffusion models for discrete states. **This result is non-trivial because it requires a careful discussion in the continuous-time Markov chain framework. In [f], they don’t have such a discussion.**
> 3) In addition, eq 12 in RTB as mentioned by the reviewer is a very general applied formulation for many RL fine-tuning papers and methods, for example, the well-known RLHF[g] and DPO[h] papers. Therefore, we do not view this as a novelty for both our paper and RTB[f]. The main focus and contribution of our paper is to solve the control problem with discrete states in the CTMC framework.
>
>
> Second, regarding the scalability concern, while we recognize the reviewer’s point, we would like to claim our method is scalable enough.
>
> 1) Direct back propagation has been used in many representative computer vision papers of fine-tuning diffusion models for image and video generation, such as [i], [j]). These papers are cited/recognized well, and in our understanding, they are treated as useful scalable methods. Like our paper, these papers use the following technique to handle large models:
>     * Truncated backpropagation like in [i].
>     * Many efficient ways: Gradient checkpointing/gradient accumulation/Lora or adapter fine-tuning.
>
> 2) The focus of this paper is on the domain of biological sequence design, which is a challenging and important application domain. Our experimental settings on enhancer design and protein inverse folding are also closely related to real-world gene therapies and protein-based therapeutics [k, l]. We focus on biological settings because we believe that masked diffusion models would shine in modeling biological sequences due to the fact that they are really discrete and non-autoregressive by nature, which fits well with masked diffusion models. Text and pixel domains are not the main focus of this paper.
>
> [f] Amortizing Intractable Inference in Diffusion Models for Vision, Language, and Control. NeurIPS 2024
>
> [g] Training language models to follow instructions with human feedback. arXiv 2022.
>
> [h] Direct Preference Optimization: Your Language Model is Secretly a Reward Model. NeurIPS 2023.
>
> [i] Directly Fine-Tuning Diffusion Models on Differentiable Rewards. ICLR 2024.
>
> [j] Video Diffusion Alignment via Reward Gradients. arXiv 2024.
>
> [k] Cell-type-directed design of synthetic enhancers. Nature 2024.
>
> [l] Aligning protein generative models with experimental fitness via direct preference optimization. bioRxiv 2024.
>
>
> > Regarding the Gumbel-Softmax estimator I would encourage the authors to try more modern gradient estimators. One suggestion is REINMAX (Liu et. al 2023). I believe this alleviates some of the bias and is more scalable than Gumbel-softmax. I also encourage the authors to do a bit more theoretical analysis of the error incurred by using as all the theory relies on not doing the Gumbel-softmax approximation so it doesn't apply to the practical settings considered in the paper.
>
>
> We thank the reviewer for the suggestions. Incorporating more advanced gradient estimators could potentially improve the model performance, and our model provides a flexible framework to allow for easy integration of future advances. While acknowledging this suggestion, it is out of the scope of our paper and we leave it as future directions.

---

> > ### Comment · Reviewer_1d7y · 2024-11-23
> > **Re: Technical concerns**
> >
> > I thank the authors again for their detailed rebuttal response.
> >
> > ### RTB Comparison
> >
> > With regards to RTB and its comparison here. Equation 12 in the RTB paper I believe matches your current setting in this paper. My claim wasn't that your setting is novel but rather that these settings are identical in formulation when we are working with Discrete Diffusion Models. The algorithms indeed differ in that RTB seeks to perform a detailed balance-based objective while here you do not need to estimate the log partition function. This is precisely why it is an ideal baseline and one that should be included. I would view this paper more favorably if the authors could manage to successfully include this baseline before the end of the rebuttal. Note that other papers have managed to successfully train RTB so I suspect it is possible and the authors should endeavor to get it to work. Alternatively, provide convincing evidence as to why RTB cannot work in this setting.
> >
> > ### Scalability
> >
> > The scalability concern persists. If we ignore the cost of doing rollouts (steps 2-9) and focus on the direct backpropagation initially then I believe it is still not scalable. Gumbel softmax is known to introduce bias and doesn't work well beyond a certain number of dimensions. Can you provide more convincing evidence that we can indeed scale to larger sequences? For instance let's say sequences of length 1024, 2048 which are quite standard input sizes.
> >
> > Now let's address the other scalability issue. The fact you need rollouts to compute a loss objective is not ideal. But this appears unavoidable in your current algorithm so perhaps we need to live with it. I will concede this point if the authors could include a timing table for their method and baselines somewhere in the paper. Even if the algorithm is expensive (or not) the paper would be stronger if we had a more exact understanding of Big O complexity + wall clock time characterization.
> >
> > ### Focus on Biological sequence design
> >
> > I disagree with the authors on this point. I believe your method is quite general and you choose to evaluate it on an important application domain in biology. Can you please explain why the current problem formulation would not work for other discrete modeling domains? What is the exact technical limitation or specific aspect of the biological problem that inhibits this?
> >
> > I find the author's suggestion that MDMs are ill-suited to other discrete generative modeling tasks a weak argument. MDMs have seen tremendous applications in text for instance. Indeed, biological sequences don't have a causal ordering and MDMs can be better suited here, but this doesn't prevent them from traditional language modeling tasks. At present, DRAKES is a very general fine-tuning method, and as such the burden of proof must be with the authors to show its potential at the very minimum in text settings. I honestly think it will work decently well here and it will make the impact of the paper stronger.

---

> ### Author Response · Authors · 2024-11-20
> **Response to Reviewer 1d7y (continued 2)**
>
> **Experimental questions**
>
> > I have some further questions and concerns regarding the considered experiments. In particular, the protein stability experiments do not indicate what length protein sequences are used. I suspect the preprocessing of PDB largely restricts the protein sequences to small ones which would be in line with my estimation that the method does not scale to larger sequences.
>
> For the protein experiment, we utilize the large-scale protein stability dataset (Megascale) that is widely used for protein optimization (eg., [l], [m]). It contains protein domains with 40-72 amino acids in length. Although the protein sequences are relatively short in length, there is not a large gap with the typical length we expect to be faced with in other protein design tasks (typically less than a few hundred, eg., [n]).
>
> The length of enhancer sequences in the enhancer experiment is 200, and we did not find the model to be unstable or hard to tune in both enhancer and protein experimental settings. We did not test DRAKES on longer protein sequences due to the lack of relevant large-scale datasets. However, given the superior performance of DRAKES on the current settings and its stability, we would expect it to be able to easily scale to protein sequences with a few hundred amino acids (a typical scale for protein design).
>
> [l] Aligning protein generative models with experimental fitness via direct preference optimization. bioRxiv 2024.
>
> [m] Unlocking Guidance for Discrete State-Space Diffusion and Flow Models. arXiv 2024.
>
> [n] De novo design of protein structure and function with RFdiffusion. Nature 2023.
>
>
> > Moreover, it is surprising why pLDDT is not reported as a metric. This is the most common metric to show that protein sequences are designable. Self-consistency makes more sense in settings where structure is generated.
>
> As shown in Table 2, self-consistency RMSD (scRMSD), which is a self-consistency measurement broadly used in previous papers (eg., [m], [o], [p], [q]), is calculated and reported.
>
> pLDDT is a similar metric to scRMSD. We additionally report the results with pLDDT in Table 12 in Appendix E.3 in the updated manuscript. The results align well with those in Table 2 using scRMSD, and DRAKES significantly outperforms all baseline methods in terms of the overall success rate.
>
> | Method | Success Rate (%)|
> | -------- | ------- |
> | Pretrained | 33.8(0.8)|
> | CG | 34.3(0.5)|
> | SMC | 53.7(4.3) |
> | TDS | 54.4(2.8) |
> | CFG | 0.0(0.0)|
> | DRAKES w/o KL | 40.4(0.2) |
> | DRAKES | **72.3(0.8)** |
>
> [m] Unlocking Guidance for Discrete State-Space Diffusion and Flow Models. arXiv 2024.
>
> [o] Generative Flows on Discrete State-Spaces: Enabling Multimodal Flows with Applications to Protein Co-Design. ICML 2024.
>
> [p] Diffusion probabilistic modeling of protein backbones in 3D for the motif-scaffolding problem. ICLR 2023.
>
> [q] An all-atom protein generative model. PNAS 2024.
>
>
>
> > Moreover, I would like to see a few more metrics that characterize the diversity of generated samples pre and post fine-tuning with DRAKES for protein sequences. Something like the number of clusters as I suspect there is likely a sharp decline in diversity which of course would not be captured by self-consistency.
>
> Thank the reviewer for the suggestion. We evaluate the diversity of the protein sequences generated by different methods using the average sequence entropy of each protein backbone in the test set. The results are shown in Table 13 of the updated manuscript. DRAKES achieves a comparable level of diversity as the pretrained model, significantly outperforming SMC and TDS. (For CG, although it also has similar diversity as the pretrained model, its success rate is very low.) This further justifies the efficacy of DRAKES to achieve the optimization objective of generating stable sequences, while maintaining high diversity. Actually, DRAKES is **the only method** among all baselines that have both a high success rate and high diversity.
>
> | Method | Sequence Entropy |
> | -------- | ------- |
> | Pretrained | 34.7(0.2) |
> | CG | 34.6(0.1)|
> | SMC | 24.9(1.2) |
> | TDS | 24.9(0.5) |
> | CFG | 8.4(0.1)|
> | DRAKES w/o KL | 25.7(0.1) |
> | DRAKES | 33.3(0.2) |
>
> Given the time constraint, we will incorporate other diversity metrics (eg., clustering-based measures or embedding-based similarity measures) later.

---

> > ### Comment · Reviewer_1d7y · 2024-11-23
> > **Re: Experimental question**
> >
> > Thank you for adding the requested metrics of pLDDT and characterization of diversity.
> >
> > I am confused by how the sequence entropy is calculated in your table above. For protein sequences with 20 amino acids the sequence entropy of a completely random sequence $H = 4.32$ bits. How is it possible the numbers above are higher than this? Can you hold my hand here, please?

---

> ### Author Response · Authors · 2024-11-20
> **Response to Reviewer 1d7y (continued 3)**
>
> > At present, DRAKES is presented as a general-purpose fine-tuning approach to discrete diffusion models but is only tested in biological sequence design. Nothing about DRAKES suggests it would not be feasible to attempt more standard discrete generative modeling settings. As such, I would like to see some standard text benchmarks for example fine-tuning for sentiment as done in Twisted SMC (Zhao et. al 2024) or class conditional fine-tuning in pixel-level image modeling. These settings would help understand both quantitatively and qualitatively, through generated samples, the empirical caliber of DRAKES.
>
> The paper is motivated by and focuses on the practical applications of biological sequence design problems, as emphasized in the abstract and introduction sections. Therefore, experimenting with text and pixel-level image benchmarks is out of the scope of this study. The enhancer sequence design and protein inverse folding are essential and standard settings in the domain of biological sequence design, and we provide rigorous and thorough evaluations on these benchmarks, with both quantitative and qualitative (eg. Figure 2 for protein inverse folding) measurements.
>
> Additionally, we want to emphasize that biological sequences are the unique domains where masked diffusion models really shine because it is inherently non-autoregressive (unlike text) and discrete (unlike pixels). Indeed, the widely successful foundation model ESM3 in this field shares similarities to masked diffusion models in both training and inference (i.e., masked iterative decoding) procedures.
>
>
>
> > Given the suspected similarity of DRAKES and RTB I would encourage the authors to include RTB as a baseline in their experimental settings.
>
> We referred to the public code of RTB, implemented it in the enhancer design setting, and tried various hyperparameters. However, with significant effort, we have not been able to make RTB work in the enhancer dataset (the average reward of the fine-tuned model stays at the same level as the pretrained model). This may be due to the difficulty in setting an exploration strategy and achieving a good estimation of the normalizing constant term. Additionally, this indicates that RTB would require careful tuning and specific techniques for optimization. In comparison, our method is stable and easy to tune and adapt, as we don't have special hyperparameters other than truncation number and Gumbel Softmax temperature, and we can just easily perform optimization with off-the-shelf optimizers in supervised learning.
>
> Given the time constraint, we will try our best to make it work by the end of the rebuttal period and post an update. We have also updated our code in the anonymous link https://anonymous.4open.science/r/DRAKES-code-5B62/, with RTB implemented in the file rebuttal_finetune_rtb.py. We would appreciate it if the reviewer had any suggestions or tricks in their mind that could make RTB work in our case.

---

> ### Author Response · Authors · 2024-11-20
> **Response to Reviewer 1d7y (continued 4)**
>
> **Presentation weakness**
>
> > All of the theory in the paper is rather incomplete and just plain sloppy. The appendices do not have complete proofs or the exact theorem statements. There is an air of informality that is scary because I believe some of the statements are actually not totally accurate and need a few assumptions as outlined previously. Lemma 1 & 2 proofs are not complete. You cannot just skip the converse part. This is not a complete proof and is a bit sloppy. I encourage the authors to complete the proof.
>
> We believe it is unfair to claim that omitting the proof of well-known results such as Komorogolow forward and backward equations is our problem. This stuff can be found on Wikipedia. https://en.wikipedia.org/wiki/Kolmogorov_equations.  In our paper, note we explicitly mentioned in Appendix B.1: ***We prepare two well-known theorems (lemma 1 and 2) in CTMC for the pedagogic purpose. For example, refer to Yin and Zhang (2012) for more detailed proof.*** We have added further clarifications in the revised paper.
>
> A well-known paper on discrete diffusion models (Page 16, [r])  and continuous diffusion models (Section D.1 in [s]) also invoke Kolmogorov’s forward equation to show their main statements. **However, they just invoke it without proving it.**
>
> While we recognize we need certain conditions to invoke Kolmogorov equations, we don’t think readers will benefit from adding detailed regularity conditions. **For example, the well-known abovementioned paper ([s]) invokes Kolmogorov forward equations in Section D.1, without mentioning regularity conditions.** Note that to the best of our knowledge, most papers in ML conferences don’t mention these regularity assumptions.
>
> [r] A continuous time framework for discrete denoising models. NeurIPS 2022.
>
> [s] Score-based generative modeling through stochastic differential equations. ICLR 2021.
>
>
> > Log Likelihood in Table 1 is actually misleading. You are likely computing an upper bound to Perplexity as done in MDLM.
>
> In our paper we provided a detailed explanation of the calculation of Log-Lik in the “Evaluation” part of Section 6.2, stating that it is calculated using the ELBO as in MDLM. We have changed the abbreviation in Table 1 to “App-Log-Lik” (representing approximated log-likelihood) in the updated manuscript, highlighted in magenta, to further make it more clear.
>
>
> **Typos and minor details**
>
> > Line 848 "competed" -> completed Concrete distribution (Maddison et. al 2016) should be cited for Gumbel-softmax as well.
>
> We thank the reviewer for pointing this out. We have corrected the typo and added the citation in the updated manuscript.

---

> > ### Comment · Reviewer_1d7y · 2024-11-23
> > **Re: Presentation weaknesses**
> >
> > The presentation weaknesses largely remain. I won't hammer this point more. I encourage the authors to take my suggestions into account if they so choose.
> >
> >
> > ### Concern about log-likelihood
> >
> > Thank you for clarifying what you are actually calculating. It is incorrect to call this approximate log-likelihood. No one calls an ELBO an approximate log-likelihood. Why is there reluctance to actually use the same terminology as MDLM (Sahoo et. al 2024) and say its an upper bound to the perplexity? This seems like a simple change to make

---

> ### Author Response · Authors · 2024-11-27
> **Response to the Latest Comments by Reviewer 1d7y**
>
> We thank the reviewer for their time and feedback. We provide our response to each additional questions below.
>
> * **Theoretical Question**
>
> We have included a discussion of the initial value bias problem and an extension of our method with stochastic initial distributions in Appendix C.4, Remark 4, and Appendix D in the updated manuscript.
>
> * **RTB Comparison**
>
> As suggested by reviewer X1rg, we run experiments with the updated code. (Note that regarding the second point mentioned by reviewer X1rg, we might be missing something but it appears to be a bug in RTB’s initial source code: https://github.com/GFNOrg/diffusion-finetuning/blob/main/diffusion_lm/finetune_cond_data.py#L154.)
>
> However, we still cannot achieve good performance (the maximal average reward it can achieve is around 0.7, which is slightly higher than the pretrained model (~0.3), while significantly lower than all other baselines).
>
> **Finally, we would also like to remind that we have already included many relevant baselines, such as CG, SMC, TDS, and CFG.**
>
> * **Scalability and Focus on Biological Sequence Design**
>
> We would like to emphasize that our experiments are not toy in the context of biological sequence design. Both the DNA dataset [5] and the protein dataset [6] are from a paper in Nature. We just don’t aim to generate randomly longer sequences without any scientific meaning.
>
> Further, analogs of our methods are known to be scalable with standard memory-efficient deep learning techniques, as shown in several representative papers ([1] for direct backpropagation and [2,3] for Gumbel Softmax).
>
> Meanwhile, we have clearly mentioned in our title, abstract, introduction, and experiment sections that the paper focuses on biological applications, where biological sequence design is an essential and challenging real-world application, and mask diffusion models are really well-suited to this setting. While acknowledging the potential applications of our method to other domains (text/pixel), this is out of the scope of our paper.
>
> * **Sequence Entropy Calculation**
>
> We calculate the sequence entropy as the summation of the entropy of each token for each sequence, following [4], and average across all generated sequences. Therefore, the maximum possible value is 4.32 * number of tokens. Calculating by averaging the entropy of each token leads to similar results.
>
> [1] Directly Fine-Tuning Diffusion Models on Differentiable Rewards. ICLR 2024.
>
> [2] Point-BERT: Pre-training 3D Point Cloud Transformers with Masked Point Modeling. CVPR 2022.
>
> [3] BEIT: BERT Pre-Training of Image Transformers. ICLR 2022.
>
> [4] Steering Masked Discrete Diffusion Models via Discrete Denoising Posterior Prediction. arXiv 2024.
>
> [5] Machine-guided design of cell-type-targeting cis-regulatory elements. Nature 2024.
>
> [6] Mega-scale experimental analysis of protein folding stability in biology and design. Nature 2023.

---

> > ### Comment · Reviewer_1d7y · 2024-11-28
> > **Re: Response**
> >
> > Thank you for attempting the RTB baseline as well as clarifying how the sequence entropy is calculated. In light of this I have increased my score from 3->5. I am opening to increasing again if the authors improve the theoretical presentation of their work to be more tailed to discrete diffusion (Theorem 2-4) plus their proofs. In addition I would like the authors to at least attempt text or give a more convincing reason why DRAKES is not well suited for that domain. If it is a latter a discussion should be added. Given that PDF can no longer be updated I would appreciate the authors writing down the statements in the comments below.

---

> ### Author Response · Authors · 2024-12-02
> **Response to the Latest Comment by Reviewer 1d7y**
>
> We thank the reviewer for dedicating their time and reassessing our work. Regarding the additional questions on improving theoretical presentation and attempting text experiments, we provide our detailed response below.
>
> **1. Theoretical Presentation**
>
> Thanks for raising this point. We will add more interpretations to highlight the difference between continuous diffusion models.
>
> * Theorem 2: This is special in discrete diffusion models. We will emphasize in the optimal control in discrete diffusion, *the ratio* is **multiplied**, while in continuous diffusion, *the score* is **added**.
>
> * Theorem 3-4: Here, we mention we use Kolmogorov forward and backward equations. To make the difference between continuous diffusion more clear, we plan to add actual forms of these equations in discrete diffusion and how they are satisfied briefly.
>
> **2. Text Experiment**
>
> We experiment on the text dataset to fine-tune a pretrained discrete diffusion model for the text infilling task. Our preliminary results show improved reward of the model during the fine-tuning process, along with improvement on other metrics (eg., GLEU, BLEU). Given the time constraint, we will update more comprehensive results in the future version of our manuscript.
>
> Nevertheless, we would like to emphasize the main focus of our paper on important biological applications, which are essential to real-world therapeutics. We experiment on biologically meaningful settings for different types of biological sequences (DNA/protein) and conduct thorough evaluations across a variety of methods, which aligns well with the motivation and focus of our paper and demonstrates the effectiveness of our method. Besides, biological sequences are different in nature from text in multiple aspects, including the non-autoregressive property, a smaller vocabulary size, and different patterns of the reward landscapes. These factors could affect the design choices of both the pretrained generative models (eg., the non-autoregressive masked discrete diffusion is more preferred) and the fine-tuning techniques.
>
> In addition, there are many previous papers (eg., [1-5]) focusing on applications in specific domains, although the proposed methods are technically general and could be potentially applicable to other domains.
>
> [1] Diffusion Probabilistic Modeling of Protein Backbones in 3D for the Motif-Scaffolding Problem. ICLR 2023.
>
> [2] The Schrödinger Bridge between Gaussian Measures has a Closed Form. AISTATS 2023.
>
> [3] Protein Design with Guided Discrete Diffusion. NeurIPS 2023.
>
> [4] Protein Discovery with Discrete Walk-Jump Sampling. ICLR 2024.
>
> [5] Dirichlet Flow Matching with Applications to DNA Sequence Design. ICML 2024.

---

> > ### Comment · Reviewer_1d7y · 2024-12-02
> > **Re: response**
> >
> > Dear authors,
> >
> > Thank you for your attempt at answering my question. I am still willing to upgrade my score. However, for this, I need to see the full modified theorem statement in the comments so that they explicitly encode the nature of the discrete diffusion setting you consider. Would you be able to provide this as requested kindly?
> >
> > Regarding the text experiment, can you please share your most current insights/findings that you have? I believe the community will benefit from a discussion on this point. With respect to upgrading the manuscript in the future, while this is a welcome decision there is no enforceable obligation which makes it hard to see if it will be done. As a result, if you could provide some discussion on this point here in a comment beyond saying that your focus is on biological applications I would be satisfied.
> >
> > Again, I wish to reiterate **I am open to upgrading my score** but I need to see the authors engage with my requests a bit more earnestly.

---

> ### Author Response · Authors · 2024-12-02
> **Gentle reminder to respond to our rebuttal**
>
> Dear reviewer 1d7y,
>
> Since the deadline for reviewer response is today, we kindly request your feedback on our latest response about the improved theoretical presentations and text experiment.
>
> We understand you may have a busy schedule, but if you have any follow-up questions or remaining concerns, please let us know. If our response has addressed your concerns, we would greatly appreciate it if you could reconsider your score and share your feedback.
>
> Thank you for your time and we truly appreciate your insights.
>
> Best regards,
>
> Authors

---

> ### Author Response · Authors · 2024-12-03
> **Response to the Latest Comment by Reviewer 1d7y**
>
> We sincerely thank the reviewer for their valuable time and thoughtful feedback, and the ongoing engagement in the rebuttal process to help us improve our work. Below, we provide detailed responses to the questions raised.
>
> **Theoretical Presentation**
>
> We provide the full modified theorem statement explicitly encoding discrete diffusion as below. Note that here the *value function [9]* refers to the equation at the bottom of page 6 of the current version of the paper, and is defined specifically to CTMC in discrete diffusion.
>
> **Theorem 2** (Optimal generator)
>
> For $x\neq y$ ($x,y\in \mathcal{X}$), the optimal CTMC generator $Q^{\theta^{\star}}_{x,y}(t)$ can be expressed by the pretrained CTMC generator $Q^{\theta\_{\mathrm{pre}}}\_{x,y}(t)$ and the value function [9]:
>
> $$Q^{\theta^{\star}}_{x,y}(t) =  Q^{\theta\_{\mathrm{pre}}}\_{x,y}(t) \exp (\\{V\_t(y)-V\_t(x)\\}/\alpha).$$
>
> **Theorem 3** (Feynman–Kac Formula in CTMC)
>
> The value function [9] follows the Feynman–Kac formulation with respect to the pretrained CTMC distribution $P^{\theta_{\mathrm{pre}}}$ generated by $Q^{\theta\_{\mathrm{pre}}}$ as follows
>
> $$\exp(V_t(x)/\alpha ) = \mathbb{E}\_{x\_{t:T}\sim P^{\theta\_{\mathrm{pre}}}}[\exp(r(x\_T)/\alpha)|x_t=x].$$
>
> **Theorem 4** (Marginal distribution induced by the optimal generator $Q^{\theta^{\star}}(t)$)
>
> The marginal distribution at time $t$ by the CTMC trajectory in [6] with generator $Q^{\theta^{\star}}(t)$, i.e., $p^{\star}_t \in \Delta(\mathcal{X})$, satisfies
>
> $$p^{\star}_t(\cdot) \propto \exp(V_t(\cdot)/\alpha )p^{\mathrm{pre}}_t(\cdot),$$
>
> where $p^{\mathrm{pre}}_t \in \Delta(\mathcal{X})$ is a marginal distribution induced by pretrained CTMC at $t$.
>
>
> **Text Experiment**
>
> Thank the reviewer for raising the discussion. First we want to point out that we only have very few days to attempt experiments on a completely new data modality and setting. Therefore, although the current findings are to the best of our knowledge based on preliminary experiments, they could be inaccurate. Given this, here are some current insights/findings we have:
> * (1) In general, the text experiment shares similar patterns to the biological sequence experiments in our paper. We are able to run the experiments on 1 80GB NVIDIA A100 GPU with batch size 16 and gradient accumulation 8.
> * (2) Similar to the biological sequence experiments, we find gradient truncation at intermediate steps helps the optimization. Gumbel softmax works with a the same value of temperature (0.5).
> * (3) Text data has a significantly larger vocabulary size compared to biological sequences (>50k versus 4 in DNA and 21 in protein). This makes the probability distribution more sparse and adding small values (eg., 1e-8) is necessary to enable success back propagation.

---

> ### Author Response · Authors · 2024-12-03
> **Could you kindly verify if the provided clarification addresses your concerns?**
>
> Dear Reviewer 1d7y,
>
> Could you kindly verify if the clarifications we provided adequately address your concerns? Your feedback is invaluable to us, and we want to ensure we’ve addressed your comments appropriately before the reviewer response deadline in 6 hours.
>
> Thank you for your time and consideration. We truly appreciate your insights and guidance.
>
> Best regards,
>
> Authors

---

> > ### Comment · Reviewer_1d7y · 2024-12-03
> > **Ack**
> >
> > Thank you for addressing my concerns. As promised, and as a show of good scholarly faith I have increased my score for 5->6. I encourage, the  author's to update the next draft with my comments in mind and continue their show if good faith.

---

> ### Author Response · Authors · 2024-12-03
> **Thanks to Reviewer 1d7y**
>
> We sincerely appreciate the reviewer's constructive feedback and continuing involvement in the rebuttal, as well as their increased score. We are pleased to hear that our rebuttal has addressed their concerns, and we will revise and update the manuscript based on the reviewer’s comments and the discussions.

---

### Official Review · Reviewer_LsP3 · 2024-11-04

**Soundness:** 2
**Presentation:** 3
**Contribution:** 3
**Rating:** 6
**Confidence:** 2

**Summary:**

This paper presents a new fine-tuning method for discrete diffusion models. The objective is to generate high-reward yet natural sequences of DNA or RNA, where naturalness is measured by the KL divergence with the prior distribution (i.e., the pretrained model). This problem is mainly formulated with KL-constrained reinforcement learning, so they use Gumbel-softmax tricks to enable backpropagation over rewards. The performance is compared with existing RL and classifier guidance methods in DNA and RNA optimization.

**Strengths:**

1. Using Gumbel-softmax in non-differentiable reinforcement learning in training discrete diffusion model is a novel approach.

**Weaknesses:**

1. This work should establish stronger connections with probabilistic inference methods that aim to sample from unnormalized densities using objectives with KL-constrained reinforcement learning. This includes hierarchical variational inference and GFlowNets, as mentioned in recent work [1] on fine-tuning diffusion models, including the discrete case.


2. Is the KL constraint for diffusion models tractable to compute? Based on theoretical results in [2], the KL divergence is intractable due to the compositionality of the diffusion process, so we can only obtain an upper bound for the KL. Please make correction that if my understanding is wrong.


3. Why should this method be better than existing RL methods? Variational inference, REINFORCE, and Gumbel-softmax are rival approaches in many applications that enable training models in non-differentiable settings. Why does the Gumbel trick have to be the best among them?

[1] Venkatraman et al. "Amortizing Intractable Inference in Diffusion Models for Vision, Language, and Control," NeurIPS 2024

[2] Fan et al. "Reinforcement Learning for Fine-Tuning Text-to-Image Diffusion Models," NeurIPS 2023.

**Questions:**

See weaknesses above.

---

> ### Author Response · Authors · 2024-11-20
> **Response to Reviewer LsP3**
>
> We appreciate the reviewer’s feedback and we are glad to see that they recognize the novelty of our work. Below, we share our thoughts on the comments and questions, including a comparison to RTB and RL algorithms, and the calculation of the KL constraint. We have updated the manuscript with changes highlighted in magenta.
>
> > This work should establish stronger connections with probabilistic inference methods that aim to sample from unnormalized densities using objectives with KL-constrained reinforcement learning. This includes hierarchical variational inference and GFlowNets, as mentioned in recent work [1] on fine-tuning diffusion models, including the discrete case.
> [1] Venkatraman et al. "Amortizing Intractable Inference in Diffusion Models for Vision, Language, and Control," NeurIPS 2024
>
> Thank you for the suggestion. Here is a summary of the differences. We have incorporated it into the main text highlighted in magenta.
>
> * It appears that [1] uses losses from trajectory balance. On the other hand, we solve a control problem with direct backpropagation. Hence, the actual algorithms are significantly different.
> * Theoretically, our result has more additional insights tailored to discrete diffusion models. As an intermediate result to show our main claim Theorem 1, we show that the optimal generator in our control problems can derive non-trivial classifier guidance in discrete diffusion models (Theorem 2 in Section 5.1.). While [1] has a certain discussion around the connection with classifier guidance in the diffusion models for continuous states, they don’t show the above insightful result for diffusion models for discrete states. This result is non-trivial because it requires a careful discussion in the continuous-time Markov chain framework. In [1], they don’t have such a discussion.
>
> Further, empirically, we tried to experiment with RTB in our settings. We referred to the public code of RTB, implemented it in the enhancer design setting, and tried various hyperparameters. However, with significant effort, we have not been able to make RTB work in the enhancer dataset (the average reward of the fine-tuned model stays at the same level as the pretrained model). This may be due to the difficulty in setting an exploration strategy and achieving a good estimation of the normalizing constant term. Additionally, this indicates that RTB would require careful tuning and specific techniques for optimization. In comparison, our method is stable and easy to tune and adapt, as we don't have special hyperparameters other than truncation number and Gumbel Softmax temperature, and we can just easily perform optimization with off-the-shelf optimizers in supervised learning.
>
> Given the time constraint, we will try our best to make it work by the end of the rebuttal period and post an update. We have also updated our code in the anonymous link https://anonymous.4open.science/r/DRAKES-code-5B62/, with RTB implemented in the file rebuttal_finetune_rtb.py. We would appreciate it if the reviewer had any suggestions or tricks in their mind that could make RTB work in our case.

---

> ### Author Response · Authors · 2024-11-20
> **Response to Reviewer LsP3 (continued)**
>
> > Is the KL constraint for diffusion models tractable to compute? Based on theoretical results in [2], the KL divergence is intractable due to the compositionality of the diffusion process, so we can only obtain an upper bound for the KL. Please make corrections if my understanding is wrong.
> [2] Fan et al. "Reinforcement Learning for Fine-Tuning Text-to-Image Diffusion Models," NeurIPS 2023.
>
> There is no discrepancy between their statement in [2] and ours. While this part might be somewhat nuanced, which may have led to a misunderstanding, we hope the following clarification resolves any confusion.
>
> First, we know in Lemma 4.2 [2], the distribution between the pretrained model distribution and the parametrized distribution with a parameter $\theta$ at the terminal end ($t=T$ in our notation) is upper-bounded by some term (we call term $\Omega(\theta)$, defined as the right-hand side of the equation in Lemma 4.2 [2]).
>
> On the other hand, when introducing our algorithm, we start with the KL divergence of path measures from $0$ to $t=T$ induced by the pretrained models and parametrized models, which corresponds to $\Omega(\theta)$. This is **completely computable**. Proposition 2 in [a] also exactly shows it. Now, we are claiming that if $\theta$ is the optimal theta in our control problem, $\Omega(\theta)$ reduces to the KL divergence between pre-trained models and parametrized models at the terminal end (since the KL between intermediate paths reduces to zero with the optimal $\theta$). In other words, the inequality in [2] becomes equality. Hence, there is no contradiction between [2] and our paper.
>
> We also mention that other related works ([b,c]) have the same statement in the continuous-time setting for the continuous state.
>
> [a] Convergence Analysis of Discrete Diffusion Model: Exact Implementation through Uniformization. arXiv 2024.
>
> [b] Fine-tuning of continuous-time diffusion models as entropy-regularized control. arXiv 2024.
>
> [c] Adjoint matching: Fine-tuning flow and diffusion generative models with memoryless stochastic optimal control. arXiv 2024.
>
>
> > Why should this method be better than existing RL methods? Variational inference, REINFORCE, and Gumbel-softmax are rival approaches in many applications that enable training models in non-differentiable settings. Why does the Gumbel trick have to be the best among them?
>
> In comparison to existing RL algorithms, to optimize the objective in Equation 5, we utilize direct backpropagation instead of other RL algorithms. Similar approaches are also utilized in [d,e] for continuous diffusion and proved to be a simple and effective approach.
>
> In the early stage of our experimental trials, we tried applying RL algorithms (eg., PPO, reward-weighted MLE) on the enhancer dataset. However, their performance appears to be very suboptimal as we observe that the learning curve of PPO is very unstable. Therefore, we converge to use direct reward backpropagation with Gumbel Softmax, which turns out to be stable and easy to tune, and performs strongly in our experiments. A potential reason for the failure of the RL algorithms is that the reward signal for each intermediate step is very noisy, and additional tricks and designs are necessary in this setting. We leave this as future work.
>
> [d] Directly Fine-Tuning Diffusion Models on Differentiable Rewards. ICLR 2024.
>
> [e] Video Diffusion Alignment via Reward Gradients. arXiv 2024.

---

> > ### Comment · Reviewer_LsP3 · 2024-11-20
> >
> > I was satisfied with the rebuttal and increased my score to 6. I was particularly convinced by the authors' efforts in exploring various RL-like methods (e.g., PPO, RTB) and their acknowledgment of the challenges, such as suboptimal results stemming from noisy intermediate rewards—a typical credit assignment problem in long trajectory optimization. Since diffusion models also involve long trajectories and episodic rewards at the terminal, this likely explains the need for a differentiable approach. I suggest the authors revise the manuscript to better highlight the motivation behind why this method could be more effective for fine-tuning discrete diffusion compared to RL-based methods, perhaps before the camera-ready submission

---

> ### Author Response · Authors · 2024-11-21
> **Response to the Latest Comment by Reviewer LsP3**
>
> We appreciate the reviewer's constructive feedback and their increased score. We are glad our revisions have addressed most of their concerns, especially regarding the comparison with RL-like methods. We will further expand on the discussion of our method's effectiveness, highlighting the advantages of direct reward backpropagation over RL-based approaches in the context of discrete diffusion fine-tuning.

---

### Author Response · Authors · 2024-11-20
**Consolidated response to all reviewers and the AC**

We thank all the reviewers for their constructive feedback that helped us improve our paper. Based on these comments, we have conducted additional experiments and revised our manuscript, with the changes highlighted in magenta. Below we provide a brief summary of the key revisions and clarifications. Please refer to the responses to individual reviewers for more details.

**Revisions**
* We included additional metrics (pLDDT and sequence entropy) to showcase the superior performance of our method in the protein inverse folding experiment. In addition, we conducted ablation studies on the gradient truncation and Gumbel Softmax temperature schedule.
* We improved the notations and added clarifications to our problem setting and formulas.
* We included discussions on additional related work RTB[a].

**Clarifications**
* **Position of this paper (reviewer 1d7y and aA7C):** The paper is motivated by and focuses on the practical applications of biological sequence design problems, essential in real-world gene therapies and protein-based therapeutics, as emphasized in the abstract and introduction sections. We want to emphasize that biological sequences are the unique domains where masked diffusion models shine since they are really discrete (unlike image) and non-autoregressive (unlike text) by nature.
* **The initial value bias problem (reviewer 1d7y and aA7C):** We use masked diffusion models for this paper and the initial distribution is a Dirac delta distribution (completely masked state) following existing works. In this case, there is no initial value bias problem. We have updated the paper to clarify this further. Additionally, our method can be expanded to settings where the initial distribution is not a Diract delta and the initial value bias problem arises.
* **Relationship with RTB[a] (reviewer LsP3 and 1d7y):** [a] uses losses from trajectory balance, while we solve a control problem with direct backpropagation. Hence, the actual algorithms are significantly different. Also, theoretically, our result has more insights tailored to discrete diffusion models.

[a] Amortizing Intractable Inference in Diffusion Models for Vision, Language, and Control. NeurIPS 2024.

---

### Meta-Review · Area_Chair_daNJ · 2024-12-21

**Metareview:**

This paper introduces a novel fine-tuning approach for discrete diffusion models, aiming to generate sequences of DNA or RNA that achieve high rewards while maintaining naturalness. Naturalness is quantified using the KL divergence from the prior distribution represented by the pretrained model. The problem is framed as a KL-constrained reinforcement learning task, where the authors leverage Gumbel-softmax techniques to enable backpropagation through non-differentiable reward functions. A key innovation lies in applying Gumbel-softmax within the non-differentiable reinforcement learning paradigm, enabling effective training of discrete diffusion models for sequence generation.

**Additional Comments On Reviewer Discussion:**

The reviewers initially raised concerns about key baseline comparisons, theoretical results, and empirical findings. However, after a thorough and detailed response from the authors, the reviewers expressed satisfaction with the clarifications provided.

---

### Decision · Program_Chairs · 2025-01-22

Accept (Poster)